# SWAN: SGD with Normalization and Whitening Enables Stateless LLM Training

**Chao Ma** [* 1]   **Wenbo Gong** [* 1]   **Meyer Scetbon** [* 1]   **Edward Meeds** [1]

## Abstract

Adaptive optimizers such as Adam (Kingma & Ba, 2015) have been central to the success of large language models. However, they often require maintaining optimizer states throughout training, which can result in memory requirements several times greater than the model footprint. This overhead imposes constraints on scalability and computational efficiency. Stochastic Gradient Descent (SGD), in contrast, is a *stateless* optimizer, as it does not track state variables during training. Consequently, it achieves optimal memory efficiency. However, its capability in LLM training is limited (Zhao et al., 2024b). In this work, we show that pre-processing SGD using normalization and whitening in a stateless manner can achieve similar performance as Adam for LLM training, while maintaining the same memory footprint of SGD. Specifically, we show that normalization stabilizes gradient distributions, and whitening counteracts the local curvature of the loss landscape. This results in SWAN (SGD with Whitening And Normalization), a stochastic optimizer that eliminates the need to store any optimizer states. Empirically, SWAN achieves $\approx 50\%$ reduction on total end-to-end memory compared to Adam. Under the memory-efficient LLaMA training benchmark of (Zhao et al., 2024a), SWAN reaches the same evaluation perplexity using half as many tokens for 350M and 1.3B model.

## 1. Introduction

Adaptive optimizers, such as Adam and its variants (Kingma & Ba, 2015; Loshchilov & Hutter, 2019; Shazeer & Stern, 2018; Pagliardini et al., 2024; Liu et al., 2023; Zhao et al., 2024a), have been central to the success of training large language models (LLMs) (Brown et al., 2020; Touvron et al., 2023b; Dubey et al., 2024; Bi et al., 2024; Bai et al., 2023; Zhang et al., 2022). However, most adaptive optimizers for LLMs are *stateful*, meaning they require tracking and maintaining internal states. While achieving remarkable empirical success, these states introduce significant memory overhead. For instance, Adam (Kingma & Ba, 2015) – the de facto optimizer for LLM training – involves the tracking of exponential moving averages (EMAs), effectively doubling memory requirements. AdEMAMix (Pagliardini et al., 2024), an extension of Adam that achieves significant convergence speed boost, requires storing even more states, tripling the memory requirements. This overhead can be significant especially in distributed settings, where the optimizer states could consume a significant amount of the GPU memory (Dubey et al., 2024; Korthikanti et al., 2023). On the other hand, while stochastic gradient descent (SGD) is optimal in terms of memory efficiency (i.e., it is *stateless*), its capability to train LLMs is limited (Zhao et al., 2024b; Zhang et al., 2020; Kunstner et al., 2023; 2024). Therefore, a natural question arises:

***Can LLMs be trained efficiently with stateless optimizers?***

There is a growing body of research that has contributed to answering this question positively, by developing novel optimizers that reduce the memory requirements associated with tracking internal state variables, while achieving similar or even speedup boost performance compared to Adam.[1] For instance, some methods rely solely on tracking the first moment of gradients (Xu et al., 2024a; Jordan et al., 2024), while others introduce an additional one-dimensional tracking variable on top of first moments (Zhang et al., 2024b; Zhao et al., 2024c). Alternatively, approaches focusing exclusively on pre-conditioner tracking have also been proposed (Pooladzandi & Li, 2024; Li, 2017). Another line of work focuses on using low-rank approximations to store the first and second moments, thereby reducing the memory cost associated with tracking optimizer states (Lialin et al., 2023; Hao et al., 2024; Zhao et al., 2024a).

In this work, we address this question by proposing to sim-

---

[*]Equal contribution  [1]Microsoft Research. Correspondence to: Chao Ma <chao.ma@microsoft.com>, Wenbo Gong <wenbogong@microsoft.com>.

*Proceedings of the 42$^{nd}$ International Conference on Machine Learning*, Vancouver, Canada. PMLR 267, 2025. Copyright 2025 by the author(s).

---

[1]See Appendix A for detailed related work review.

ply pre-process the instantaneous stochastic gradient in a stateless manner. The result is SWAN (SGD with Whitening And Normalization), a novel stochastic optimizer that eliminates *all* internal optimizer states and empirically achieves comparable or even better performance compared to Adam on several LLM pre-training tasks. Our optimizer consists of combining two known operators to pre-process the raw gradients: `GradNorm` and `GradWhitening`. `GradNorm` applies a row-wise standardization on the gradient matrix, while `GradWhitening` orthogonalizes the normalized gradient matrix. On a theoretical model of transformer (Tian et al., 2023), we show that these operators help *stabilizing* the stochasticity of gradient distributions during training, and *mitigating* the local geometry of the loss landscape, respectively. When applied in tandem, they eliminate the need to track state variables, thereby recovering the memory footprint of SGD, while matching the performance of Adam. To summarize, our contributions are:

- **SWAN, a practical, stateless, adaptive optimizer**, provides the following properties:

    1. Memory efficiency: SWAN achieves the memory footprint of SGD, that is $\approx 50\%$ reduction on total memory, and $\approx 100\%$ reduction on optimizer states compared to Adam.
    2. Sample efficiency: with SWAN, we obtain the first evidence that stateless training of LLMs can be possible. Across several LLM scales, SWAN consistently achieves the same or better performance than Adam. On the memory-efficient LLaMA training benchmark (Zhao et al., 2024a), SWAN even improves upon the token-efficiency of Adam (2x speedup).
    3. Robustness to ill-conditioned problems and hyperparameters: on LLM pre-training, SWAN consistently and efficiently converges *without* learning rate warm-up (Section 5.1, Section 5.2) and extensive tuning.

- **Theoretical consistency with LLM dynamnics.** We show that (1) `GradNorm` can stabilize the heterogeneous covariance of LLM gradients, leveraging the redundancies in transformer gradient flows (Theorem 1 in Appendix C); and (2) `GradWhitening` can be derived as a non-diagonal second-order update under a specific structural assumption of the Hessian (Section 4.3). Additionally, we highlight that in the quadratic case, `GradWhitening` leads to convergence rates that are robust to the condition number of the local curvature (Theorem 2, Appendix C.1).

## 2. Preliminaries

Adam (Kingma & Ba, 2015) is the current standard choice for adaptive optimizers on LLM pre-training tasks. Adam is

an example of a *stateful* optimizer, which means it accumulates and stores internal states throughout training. Consider a loss function $\mathcal{L}_{\mathbf{W}} : \mathcal{X} \to \mathbb{R}$, parameterized by weight matrices $\mathbf{W} \in \mathbb{R}^{m \times n}$, and denote $\boldsymbol{x}^{(t)}$ a mini-batch of inputs provided at the $t$-th training step that is sampled from data distribution $p_{\text{data}}(\boldsymbol{x})$. Let $\mathbf{G}^{(t)}$ be the stochastic gradient of $\mathcal{L}_{\mathbf{W}}$ (i.e., a random variable induced by sampling $\boldsymbol{x}^{(t)}$). Then, Adam can be broken down into the following steps:

$$\boldsymbol{m}^{(t)} = \beta_1 \boldsymbol{m}^{(t-1)} + (1 - \beta_1)\mathbf{G}^{(t)},$$
$$\boldsymbol{\nu}^{(t)} = \beta_2 \boldsymbol{\nu}^{(t-1)} + (1 - \beta_2)\mathbf{G}^{(t)2},$$
$$\mathbf{W}^{(t+1)} = \mathbf{W}^{(t)} - \eta\Big(\frac{\hat{\boldsymbol{m}}^{(t)}}{\sqrt{\hat{\boldsymbol{\nu}}^{(t)}} + \epsilon}\Big)$$

where $\boldsymbol{m}^{(t)}$ and $\boldsymbol{\nu}^{(t)}$ are EMAs of the first and second moments of the gradients; and $\eta$ is a global step size. Tracking and storing these two EMA estimates triples the total memory consumption required to train a LLM model. The LLaMA 405B model, for example, requires an additional 1.6TB of memory for Adam compared to SGD.

**Desired properties of Adam.** There is a rich literature on understanding adaptive methods' inner workings and unreasonable effectiveness. Notably, we outline Adam's key properties that relate to this work, that have been proposed and supported by studies in the literature:

- *Gradient whitening* [3] : it is known that the inverse second moment $\frac{1}{\sqrt{\hat{\boldsymbol{\nu}}^{(t)}} + \epsilon}$ of Adam performs gradient whitening by approximating the square root inverse of the diagonal of Fisher information matrix (Kingma & Ba, 2015; Hwang, 2024), as opposed to the inverse in the standard natural gradient descent. This step biases the optimization trajectories towards well-conditioned regions (Jiang et al., 2024) and provides a better approximation to the geodesic flow when compared with the natural gradient update (Yang & Laaksonen, 2008).

- *Gradient smoothing*: the EMA operations in Adam naturally reduce the influence of mini-batch noise (Cutkosky & Mehta, 2020; Crawshaw et al., 2022);

- *Gradient invariance*: recent work suggest the performance gap between SGD and Adam might lie in Adam's *sign-descent*-like nature (Bernstein et al., 2018; Crawshaw et al., 2022; Chen et al., 2023). For example, Adam is robust to the rescaling of gradient diagonals (Kingma & Ba, 2015); and is invariant to any sign-preserving scalings (under $\beta_1 = \beta_2 = 0$) (Bernstein et al., 2018).

---

[3]"Whitening" is an over-loaded term. In this paper, we primarily refer to natural gradient descent with exponent $-\frac{1}{2}$ (Yang & Laaksonen, 2008), as opposed to the work of (Amari, 1998) (which is also referred to as whitening in some context.)

Table 1: Memory consumption breakdown of optimizers, and the corresponding learning rate expressiveness. We assume all optimizers are applied on 2D tensor of $m$ by $n$ ($m < n$). The optimizer state is defined as all intermediate variables maintained over time; expressiveness is the number of adaptive learning rates that a optimizer can express per tensor.

|  | Adam | SGD | SGD-Sal | Apollo (rank $r$) | Apollo (mini) | GaLore (rank $r$) | SWAN |
|---|---|---|---|---|---|---|---|
| Optimizer States ↓ | $2mn$ | $0$ | $mn$ | $2nr + 2 + (mr)^2$ | $2n + 2 + (m)$ | $2nr + mr$ | $0$ |
| Model weights ↓ | $mn$ | $mn$ | $mn$ | $mn$ | $mn$ | $mn$ | $mn$ |
| Expressiveness ↑ | $mn$ | $1$ | blockwise | $n$ | $1$ | $mn$ | $mn$ |

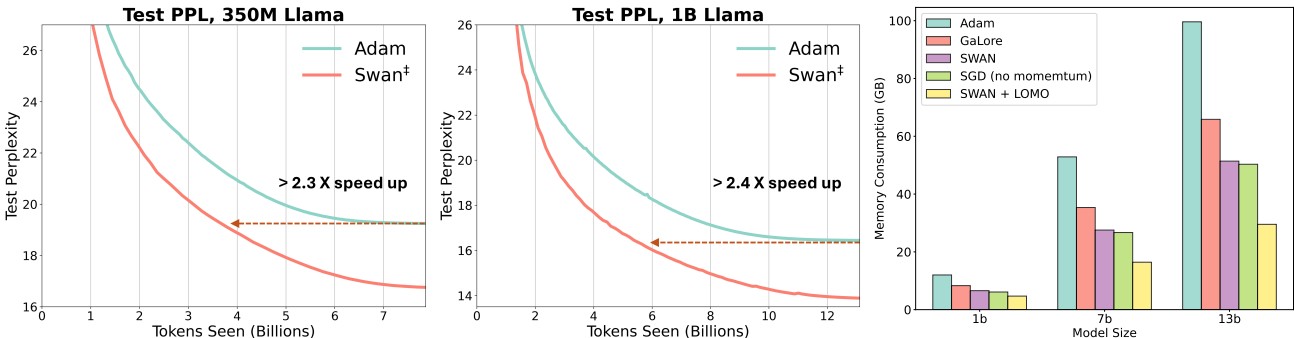

(a) 350 M LLaMA model Pretraining  (b) 1.3 B LLaMA model Pretraining  (c) End-to-end memory footprint

Figure 1: **SWAN performance preview on memory-efficient LLaMA pretraining benchmark (Zhao et al., 2024a). (a)** and **(b)**: On both 350M and 1.3B models, SWAN achieves $> 2X$ speed-up vs Adam in terms of tokens seen. **(c)**: memory footprint. We directly measure end-to-end memory under full-model training, with batch size = 1 sequence. SWAN achieves near-SGD optimizer memory reduction ($\approx 50\%$ reduction on total memory, and $\approx 100\%$ reduction on optimizer states).

For a more comprehensive discussion of these properties, please refer to appendix B.

**Adam as SGD pre-processing.** Adam can be understood as computing a history-dependent pre-processing of the gradients ($\{\mathbf{G}^{(0)}, \mathbf{G}^{(1)}, ..., \mathbf{G}^{(t)}\} \rightarrow \frac{\hat{\boldsymbol{m}}^{(t)}}{\sqrt{\hat{\boldsymbol{\nu}}^{(t)}} + \epsilon}$) to achieve the desired properties described above. A key observation is that all of these properties are achieved through element-wise operations, where each element of the gradient matrix is independently pre-processed and re-scaled. This approach does not take into account the interactions and structures between different variables, and we hypothesize that this is the reason why additional history information is necessary to bridge this gap, ultimately leading to the requirement for EMA states. We believe that designing stateless adaptive optimizers is possible if we can achieve similar properties by applying matrix-level operations that pre-process the instantaneous stochastic gradients of SGD ($\mathbf{G}^{(t)} \rightarrow \tilde{\mathbf{G}}^{(t)}$).

# 3. The SWAN Optimizer: Preprocessing SGD with Normalization and Whitening

As discussed in Section 2, [4] the key to designing stateless adaptive optimizers lies in the incorporation of *matrix-level*

operations that exploit rich information contained in the gradient matrix. To this end, we compose two well-known matrix operators, namely normalization and whitening. When applied in tandem, they achieve similar desirable properties of adaptive optimizers, without the need to store historical gradient moments. The result is SWAN (SGD with Whitening And Normalization), a new stateless optimizer.

## 3.1. SWAN Update Rules

In SWAN (Algorithm 1), the raw SGD gradient $\mathbf{G}^t$ is processed by the following operations:

$$\begin{cases} \tilde{\mathbf{G}}^{(t)} \leftarrow \texttt{GradNorm}(\mathbf{G}^{(t)}) \\ \Delta\mathbf{W}^{(t)} \leftarrow \texttt{GradWhitening}(\tilde{\mathbf{G}}^{(t)}) \end{cases} \quad \text{(SWAN)}$$

The weight is then updated by $\mathbf{W}^{(t+1)} = \mathbf{W}^{(t)} - \eta\Delta\mathbf{W}^{(t)}$. The `GradNorm` operator (Equation (1)) denotes the normalization of the gradient matrix row-wise (Figure 2); and the `GradWhitening` operator denotes the whitening of the gradients (Figure 2).

**On the `GradNorm` Step.** Consider the gradient matrix $\mathbf{G} \in \mathbb{R}^{m \times n}$ with rows and columns corresponding to input and output dimensions, respectively, of some block of model

---

[4] $mr$ comes from the storage of random projection tensors.

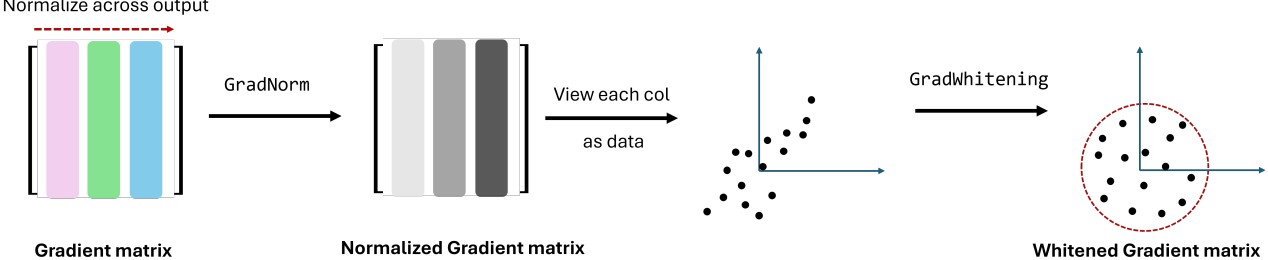

Figure 2: Illustration of `GradNorm` and `GradWhitening` operators. In `GradNorm` operator, we perform normalization across the output dimensions (columns), using statistics computed row-wise. In `GradWhitening` operator (illustration adapted from Huang et al. (2019)), we treat each column of the gradient matrix $\mathbf{G}$ as a separate data sample. Then, `GradWhitening` can be seen as squeezing the gradient data along all eigen directions, such that the covariance matrix is the identity matrix.

parameters. Let $1 \le i \le m$ represent the input indices and $1 \le j \le n$ represent the output indices. Instead of performing element-wise EMA to stabilize and normalize the noisy gradients, as done in Adam, we propose to standardize across the output dimensions at each time step $t$:

$$\texttt{GradNorm}(\mathbf{G}) := \frac{\mathbf{G}}{s\mathbf{1}_n^\top} \qquad (1)$$

where $s := \sqrt{\frac{1}{n}\sum_{j=1}^{n}(\mathbf{G}_{:,j})}$ is $m$-dimensional the root mean square across dimension ($m$-dimensional column vector); $\mathbf{1}_n$ is a $n$-dimensional column vector of ones.

`GradNorm` is the forward pass operator RMS-Norm (Zhang & Sennrich, 2019) (without learnable parameters) applied on backward gradients. Similar operations have been considered in the literature to process gradients (You et al., 2019). `GradNorm` allows the optimizer to be invariant under matrix-wise and row-wise rescaling. In Section 4 we will show that this is the key to stabilizing the LLM gradient distribution. Meanwhile, compared to the sign operation ($\text{sign}(\mathbf{G})$) and matrix-wise normalization ($\frac{\mathbf{G}}{\|\mathbf{G}\|}$) used in signed descent, `GradNorm` preserves richer information of the gradient scaling while offering invariance properties.

**On the `GradWhitening` Step.** As discussed in Section 2, Adam relies on a second moment estimate to perform *element-wise* gradient whitening. More formally, the second moment of Adam estimate a diagonal approximation of the Fisher information matrix (FIM) via (Hwang, 2024):

$$\mathbb{E}(\text{Vec}(\mathbf{G})\text{Vec}(\mathbf{G})^\top) \approx \text{Diag}[\text{Vec}[\mathbb{E}(\mathbf{G}^{\odot 2})]]$$

where $\text{Vec}(\cdot)$ denotes the vectorized operation, $\text{Diag}(\cdot)$ denotes the operation that produces a diagonal matrix given a vector, and $\odot$ denotes the Hadamard product. Hence, whitening using the inverse square root of this diagonal FIM is equivalent to element-wise rescaling with $\frac{1}{\sqrt{\mathbb{E}(\mathbf{G}\odot\mathbf{G})}}$. Here, we consider non-diagonal approximation:

$$\mathbb{E}(\text{Vec}(\mathbf{G})\text{Vec}(\mathbf{G})^\top) \approx \mathbf{I}_{n\times n} \otimes \mathbf{G}\mathbf{G}^\top \ ,$$

where $\otimes$ denotes Kronecker product. This leads to the following whitening step:

$$\texttt{GradWhitening}(\mathbf{G}) := (\mathbf{G}\mathbf{G}^\top)^{-1/2}\mathbf{G} \qquad (2)$$

where the exponent $-\frac{1}{2}$ stands for matrix inverse square root. $(\mathbf{G}\mathbf{G}^\top)^{-\frac{1}{2}}\mathbf{G}$ is simply the orthogonalization of $\mathbf{G}$, i.e. the closest orthogonal matrix to $\mathbf{G}$ (w.r.t the Frobenius norm). The derivation of (2) as structured Hessian approximation is discussed in Section 4.3. Finally, Equation (2) can also be interpreted as *minimizing the condition number of* $\mathbf{G}$.

Similar to `GradNorm`, `GradWhitening` (Equation (2)) as a matrix operation has been widely used as a forward-pass operator in the form of decorrelated batch normalization (Huang et al., 2018; Li et al., 2018); and it has also shown initial success in processing backward gradients (Tuddenham et al., 2022; Jordan et al., 2024; Gupta et al., 2018). As shown in Figure 2, by treating each column of $\mathbf{G}$ as i.i.d. vector-valued data samples $\mathbf{G} = \{\boldsymbol{g}_1, ..., \boldsymbol{g}_j, ..., \boldsymbol{g}_n\}$, `GradWhitening` can be seen as effectively stretching/squeezing this data matrix along the eigenvectors to whiten its covariance matrix. This forces the gradients to *traverse all eigen-directions at the same rate*.

### 3.2. Practical Considerations

**Rescaling Updates.** The operator `GradWhitening` maps the normalized gradient $\tilde{\mathbf{G}}^{(t)}$ onto the closest orthogonal matrix, and as such might drastically change its effective learning rate. In practice, we propose the following rescaling before updating the weights:

$$\Delta\mathbf{W}^{(t)} \leftarrow \frac{\|\tilde{\mathbf{G}}^{(t)}\|\Delta\mathbf{W}^{(t)}}{\|\Delta\mathbf{W}^{(t)}\|} = \frac{\sqrt{mn}\Delta\mathbf{W}^{(t)}}{\|\Delta\mathbf{W}^{(t)}\|} \qquad (3)$$

This helps rescale the norm of the whitened gradient back to the norm of $\tilde{\mathbf{G}}^{(t)}$, that is $\|\tilde{\mathbf{G}}^{(t)}\| = \sqrt{mn}$. After we proposed Equation (3), follow-up work of (Gong et al., 2025) showed that Equation (3) can be naturally derived

---

**Algorithm 1** SWAN Optimizer

---

**Input:** weight matrix $\mathbf{W} \in \mathbb{R}^{m \times n}$ with $m \leq n$. Step size $\eta$. Number of GradWhitening iteration $K$ (default = 10).

Initialize step $t \leftarrow 0$
**repeat**
  $\mathbf{G}^{(t)} \in \mathbb{R}^{m \times n} \leftarrow \nabla_{\mathbf{W}^{(t)}} \mathcal{L}^{(t)}(\mathbf{W}^{(t)})$
  $\tilde{\mathbf{G}}^{(t)} \leftarrow \texttt{GradNorm}(\mathbf{G}^{(t)})$
  $\Delta\mathbf{W}^{(t)} \leftarrow \texttt{GradWhitening}(\tilde{\mathbf{G}}^{(t)}, K)$
  (optional) $\Delta\mathbf{W}^{(t)} \leftarrow \frac{\sqrt{mn}\Delta\mathbf{W}^{(t)}}{\|\Delta\mathbf{W}^{(t)}\|}$
  $\mathbf{W}^{(t)} \leftarrow \mathbf{W}^{(t-1)} - \eta\Delta\mathbf{W}^{(t-1)}$
  $t \leftarrow t + 1$
**until** convergence criteria met
**return** $\mathbf{W}^{(t)}$

---

**Algorithm 2** GradWhitening Operator

---

**Input:** $\mathbf{G}^{m \times n}$ with $m \leq n$. Number of iterations $K$. Step size $\beta$. Boolean diag indicating if use diagonal substitution scheme.
Initialize $\mathbf{Y} \leftarrow \mathbf{GG}^\top, \mathbf{Z} \leftarrow \mathbf{I}$
**for** i = 1,...,K **do**
  **if** diag **then**
    $\mathbf{Y} \leftarrow \beta\mathbf{Y}\text{Diag}(3\mathbf{I} - \mathbf{Z}\text{Diag}(\mathbf{Y}))$,
    $\mathbf{Z} \leftarrow \beta(3\mathbf{I} - \text{Diag}(\mathbf{Z})\mathbf{Y})\text{Diag}(\mathbf{Z})$
  **else**
    $\mathbf{Y} \leftarrow \beta\mathbf{Y}(3\mathbf{I} - \mathbf{ZY})$,
    $\mathbf{Z} \leftarrow \beta(3\mathbf{I} - \mathbf{ZY})\mathbf{Z}$
  **end if**
**end for**
**return** $\mathbf{ZG}$

---

**Algorithm 3** GradNorm Operator

---

**Input:** $\mathbf{G}^{m \times n}$
**return** $\frac{\mathbf{G}}{\sqrt{\frac{1}{n}\sum_{j=1}^{n}(\mathbf{G}_{:,j})^2}\mathbf{1}_n^\top}$

---

Figure 3: SWAN Optimizer.

from a unifying framework for LLM optimization based on Fisher information. Later, the same scaling (noticing $\|\Delta\mathbf{W}^{(t)}\| = \sqrt{m}$ ) has been used in (Liu et al., 2025) to apply whitening operations to larger models.

**Fast GradWhitening** Exact GradWhitening is expensive, as it involves solving the matrix square-root inverse. We can instead directly apply the Newton-Schulz (NS) variant of decorrelated batch normalization (Song et al., 2022; Li et al., 2018; Huang et al., 2019), which allows a more

GPU-friendly estimation. This is given by:

$$
\begin{cases}
\mathbf{Y}_{k+1} = \frac{1}{2}\mathbf{Y}_k(3\mathbf{I} - \mathbf{Z}_k\mathbf{Y}_k)) \\
\mathbf{Z}_{k+1} = \frac{1}{2}(3\mathbf{I} - \mathbf{Z}_k\mathbf{Y}_k)\mathbf{Z}_k
\end{cases}
\tag{4}
$$

where $\mathbf{Y}_0 = \mathbf{GG}^\top$, $\mathbf{Z}_0 = \mathbf{I}$. At convergence, $\mathbf{Z} = (\mathbf{GG}^\top)^{-1/2}$, hence $\texttt{GradWhitening}(\mathbf{G}) = \mathbf{ZG}$ (Algorithm 2). See Appendix K for implementation details. The use of the same N-S procedure for Shampoo variants has been discussed in (Mei et al., 2023) and (Jackson, 2023). Another N-S procedure was applied in (Bernstein & Newhouse, 2024b), and optimized in (Jordan et al., 2024). However, estimating $(\mathbf{GG}^\top)^{-1/2}$ with NS requires $\mathcal{O}(m^3)$ (assuming $m < n$) complexity, making its scalability for larger models, especially under certain frontier settings unclear (Essential AI, 2025). Here, we propose a heuristic scheme that has $\mathcal{O}(m^2)$ complexity:

$$
\begin{cases}
\mathbf{Y}_{k+1} = \frac{1}{2}\mathbf{Y}_k\text{Diag}(3\mathbf{I} - \mathbf{Z}_k\text{Diag}(\mathbf{Y}_k))) \\
\mathbf{Z}_{k+1} = \frac{1}{2}(3\mathbf{I} - \text{Diag}(\mathbf{Z}_k)\mathbf{Y}_k)\text{Diag}(\mathbf{Z}_k)
\end{cases}
\tag{5}
$$

where $\text{Diag}(\cdot)$ returns a diagonal matrix that has the same diagonal elements as the input matrix. Basically, whenever we encounter matrix multiplication in NS iterations, we replace one of them by its diagonal approximation. We refer to this as the *NS with diagonal substitution* (NSDS) scheme. In Appendix I, we empirically demonstrate that NSDS performs well in minimizing the matrix condition number of gradients. As shown in Figure 6, on the gradient distribution induced by Llama model training, the performance of NSDS (when combined with GradNorm) is comparable to the standard NS scheme in reducing the condition number.

We found that NSDS does have the side effect of significantly slower early stage convergence speed. However, it offer stronger tail convergence in return. For LLM pretraining NSDS helps SWAN achieve similar or better final validation loss performance than the original NS scheme (Section 5.1), enabling Adam-level training throughput without the help of sharding NS computation (Section 5.3).

## 4. Why SWAN: A LLM Dynamics Perspective

As a first analysis, we consider SWAN from a *learning dynamics* perspective, specifically the dynamics of an LLM based upon a simplified transformer block. It is this analysis that led to the design choices for SWAN. Theoretical insights of this section is verified empirically in Appendix J.1.

### 4.1. Setup

We consider the simplified transformer block (STB) architecture proposed in (Tian et al., 2023).

**Definition 1** (Simplified Transformer Block). *Given the input $\boldsymbol{x} \in \mathbb{R}^{M_C \times 1}$, query $q$, context embedding $\mathbf{U}_C \in \mathbb{R}^{d \times M_C}$, and the query embedding*

$\boldsymbol{u}_q \in \mathbb{R}^{d \times 1}$, *the STB computes the output as* $\boldsymbol{h} = \phi \left( \mathbf{W}^\top \left( \mathbf{U}_C \left( \exp(\boldsymbol{z}_q) \odot \boldsymbol{x} \right) + \boldsymbol{u}_q \right) \right)$, *where* $M_C$ *is the context length. The attention logits* $\boldsymbol{z}_q \in \mathbb{R}^{M_C \times 1}$ *are given by* $z_{ql} = \boldsymbol{u}_q^\top \mathbf{W}_Q^\top \mathbf{W}_K u_l$, *with* $\mathbf{W}_Q, \mathbf{W}_K \in \mathbb{R}^{d \times d}$ *being weights for queries and keys, respectively,* $\mathbf{W} \in \mathbb{R}^{d \times n}$ *is the MLP project-up weights, and* $\phi$ *is a nonlinear function.*

Given a STB, we consider a loss function $\mathcal{L}_{\mathbf{W},\boldsymbol{z}}(\boldsymbol{x}^{(t)})$, where $\boldsymbol{x}^{(t)}$ is a mini-batch of inputs provided at the $t$-th training step sampled from data distribution $p_{\text{data}}(\boldsymbol{x})$. Standard mini-batch learning dynamics is then given by

$$\dot{\mathbf{W}}^{(t)} = \frac{\partial \mathcal{L}_{\mathbf{W},z_q}(\boldsymbol{x}^{(t)})}{\partial \mathbf{W}}, \quad \dot{z}_q^{(t)} = \frac{\partial \mathcal{L}_{\mathbf{W},z_q}(\boldsymbol{x}^{(t)})}{\partial \boldsymbol{z}_q} .$$

In this case, both $\dot{\mathbf{W}}^{(t)}$ and $\dot{z}_q$ are viewed as random variables induced by random mini-batch $\boldsymbol{x}^{(t)}$. For example, for each row $i$, $\dot{\mathbf{W}}^{(t)}[i, :]$ can be re-written as $\dot{\mathbf{W}}^{(t)}[i, :] = \mathbb{E}[\dot{\mathbf{W}}^{(t)}[i, :]] + \boldsymbol{\varepsilon}_{\mathbf{W}}^{(t)}[i, :]$, where $\boldsymbol{\varepsilon}^{(t)}[i, :]$ is zero mean random variable with covariance $\text{Cov}[\dot{\mathbf{W}}^{(t)}[i, :]]$.

### 4.2. `GradNorm`: stabilizing gradient distributions of STB

Below we show that, based on the dynamics of the STB, `GradNorm` stabilizes $\boldsymbol{\varepsilon}_{\mathbf{W}}^{(t)}$.

**Theorem 1** (`GradNorm` **stabilizes gradient distributions across time** for the STB). *Consider the STB (Definition 1). Assuming we inherit the assumptions in Theorem 1 of Tian et al. (2023), as described in Appendix C. Then consider* $\mathbf{U}_C^\top \mathbf{W}$, *the composition of the MLP project-up matrix and the embedding matrix as a whole. Then, its standardized stochastic gradients* $\tilde{\mathbf{G}}_{\mathbf{U}_C^\top \mathbf{W}}^{(t)} := \texttt{GradNorm}(\frac{\partial \mathcal{L}_{\mathbf{W},\boldsymbol{z}}(\boldsymbol{x}^{(t)})}{\mathbf{U}_C^{\mathbf{W}}})$ *satisfy:*

$$\text{Cov}[\tilde{\mathbf{G}}_{\mathbf{U}_C^\top \mathbf{W}}[i, :]^{(t_1)}] = \text{Cov}[\tilde{\mathbf{G}}_{\mathbf{U}_C^\top \mathbf{W}}[i, :]^{(t_2)}]$$

*For all* $t_1, t_2,$ *and* $i$. *In other words, the covariance structure of* $\tilde{\mathbf{G}}$ *is identical across all time steps* $t$, *achieving distributional stability across time. The same relationship also holds for the gradient of attention score* $\tilde{\mathbf{G}}_{\boldsymbol{z}_q}^{(t)} :=$ `GradNorm`$(\frac{\partial \mathcal{L}_{\mathbf{W},\boldsymbol{z}_q}(\boldsymbol{x}^{(t)})}{\partial \boldsymbol{z}_q})$.

Theorem 1 suggests that `GradNorm` implicitly aligns with the dynamics of transformer architectures and removes the time-heterogeneity in gradient covariance structures.

### 4.3. `GradWhitening`: exploiting the tied block-diagonal structures of transformer Hessian

Here we show that `GradWhitening` is equivalent to a non-diagonal second-order method under a specific Kronecker factorization assumption of the Hessian/FIM.

**Assumption 1** (Assumption of `GradWhitening`). *At time* $t$, *the local Hessian* $\mathbf{H}$ *of the loss has shared block-diagonal structure* $\mathbf{H} = \mathbf{I}_{n \times n} \otimes \tilde{\mathbf{H}}$, *where* $\tilde{\mathbf{H}} \in \mathbb{R}^{m \times m}$, $m < n$.

Approximating Hessian with a Kronecker factorization is not new and has been extensively studied in the literature (Martens & Grosse, 2015; George et al., 2018; Gao et al., 2023; 2021; Koroko et al., 2022; Eschenhagen et al., 2024; Gupta et al., 2018). This specific structure is useful in our context because it aligns with the statistical property of STB, as shown later in Proposition 1.

By leveraging Assumption 1, we can now effectively estimate $\mathbf{H}$ by only using one single gradient matrix sample $\boldsymbol{G} := [\boldsymbol{g}_1, \ldots, \boldsymbol{g}_n] \in \mathbb{R}^{m \times n}$. Recall that the Fisher information formulation of Hessian is defined as $\mathbf{H} = \mathbb{E}[\text{Vec}(\mathbf{G})\text{Vec}(\mathbf{G})^\top]$ where $\text{Vec}(\cdot)$ denotes the vectorized operation. Under assumption 1, we can estimate $\mathbf{H} = \mathbf{I}_{n \times n} \otimes \tilde{\mathbf{H}}$ by $\tilde{\mathbf{H}} \approx \frac{1}{n} \sum_{i=1}^n \boldsymbol{g}_i \boldsymbol{g}_i^\top = \frac{1}{n} \boldsymbol{G} \boldsymbol{G}^\top$. In fact, in the follow-up work of (Gong et al., 2025) it was formally showed that $\frac{1}{n} \mathbf{G} \mathbf{G}^\top$ is the solution of

$$\min_{\mathbf{M}} \| \mathbf{I}_{n \times n} \otimes \mathbf{M}_{m \times m} - \mathbb{E}[\text{Vec}(\mathbf{G})\text{Vec}(\mathbf{G})^\top] \|_F^2 \quad (6)$$

Hence, `GradWhitening` can be seen as applying a second order update [5] under our structural assumption:

$$\texttt{GradWhitening}(\mathbf{G}) = \sqrt{\max(m, n)} (\mathbf{G}\mathbf{G}^\top)^{-\frac{1}{2}} \mathbf{G}$$

Noticing $\|\Delta \mathbf{W}^{(t)}\| = \sqrt{m}$, we recovered the rescaled whitening of eq. (3). In the following Proposition, we show that the assumption 1 of `GradWhitening` aligns with the equilibrium Hessian structure in the STB regime.

**Proposition 1** (**Shared structures in the block-diagonal of Hessians at transformer equilibrium**). *Consider a STB (1), trained with full-batch gradient descent. Next, assume we inherit all the assumptions from Theorem 1 of (Tian et al., 2023). Then, as* $t \to \infty$, *we have the following shared Hessian structure along the diagonal blocks:*

$$\frac{\mathbf{H}_{sk,s'k}}{\sum_{s,s'} \mathbf{H}_{sk,s'k}} \to \frac{\mathbf{H}_{sk',s'k'}}{\sum_{s,s'} \mathbf{H}_{sk',s'k'}} \quad (7)$$

*For all* $1 \leq s, s' \leq d, 1 \leq k, k' \leq n$, *Where* $\mathbf{H}(\mathbf{W})_{sk,s'k'} = \frac{\partial \mathcal{L}}{\partial w_{sk} \partial w_{s'k'}}$.

Proposition 1 shows that, under a simplified setting of the transformer, the Hessian will also converge to an equilibrium solution where the $M_C \times M_C$ blocks over the diagonal direction of Hessian shares an identical structure, which supports the assumption of `GradWhitening`. We verify this experimentally (Appendix, Figure 5). Furthermore, Appendix Theorem 2 shows that `GradWhitening` makes SGD's convergence more robust to local curvature conditions and outperforms both SGD and Adam in ill-conditioned scenarios.

---

[5] Using the natural gradient update rule with exponent $-\frac{1}{2}$ (Gong et al., 2025; Yang & Laaksonen, 2008) $\text{Vec}(\mathbf{W}^{(t+1)}) - \text{Vec}(\mathbf{W}^t) \propto -\mathbb{E}[\text{Vec}(\mathbf{G})\text{Vec}(\mathbf{G})^\top]^{-1/2}\text{Vec}(\mathbf{G})$

# 5. Experiments[6]

## 5.1. SWAN Performance on LLM Pre-training Tasks

**Setup**   We evaluate SWAN on memory-efficient Llama (Touvron et al., 2023a) pre-training tasks, using the standardized settings of (Zhao et al., 2024a), which has been adopted by many recent works (Zhao et al., 2024a; Chen et al., 2024; Zhao et al., 2024a; Zhu et al., 2024). We consider models with 60M, 130M, 350M, and 1.3B parameters, all trained on the C4 dataset (Raffel et al., 2020) using an effective batch size of 130K tokens. Following the setup of (Zhao et al., 2024a), SWAN is applied to all linear modules in both attention and MLP blocks. Training uses BF16 by default unless specified, see Appendix K. The other evaluation settings follows Zhao et al. (2024a).

**SWAN optimizer**   We consider three variants: **SWAN-0**, which aims to show-case the robustness and effectiveness of our method out-of-the-box, with almost no tuning. It uses naive NS-iteration for whitening, disabled learning rate warmup, and use similar learning rates optimized for Adam. **SWAN†**, the vanilla version of our method, in which we enabled learning rate warmup, and allowed the use of optimized learning rates that largely differ from Adam. Finally, **SWAN‡**, the strongest and most efficient variant that employs the proposed NSDS scheme for fast whitening (section 3.2). For all variants, we adopt a *lazy-tuning approach* (hyperparameters are set without extensive search) to reduce the possibility of unfair performance distortion. Notably, for SWAN‡, we share the same hyperparameters across all model sizes. Full details can be found in Appendix K.

**Baselines**   We compare SWAN with **Adam** (Kingma & Ba, 2015) and other memory-efficient optimizers. All baseline results are directly quoted from corresponding papers as they all share the same standard setup. The baselines include **Adam** (Kingma & Ba, 2015), which is a standard choice for training large models; and **Galore** (Zhao et al., 2024a), a memory-efficient Adam variant with low-rank gradient projections. We also consider **Low-Rank** (Kamalakara et al., 2022), a low-rank factorization approach ($\mathbf{W} = \mathbf{BA}$); and **LoRA** (Hu et al., 2021), which applies the LoRA method for pre-training as in (Zhao et al., 2024a). Additionally, we include **ReLoRA** (Lialin et al., 2023), a full-rank extension of LoRA with parameter merging, and **Momentum**+`GradWhitening`, which applies Newton-Schulz `GradWhitening` on top of momentum instead of `GradNorm`; this is similar to Muon (Jordan et al., 2024) with Nesterov acceleration turned off. Finally, we compare with **Apollo-mini** and **Apollo** (Zhu et al., 2024). Full details can be found in Appendix K.

---

[6]Code will be released at github.com/microsoft/msr_optim

**Results**   As shown in Table 2 and Figure 1, all SWAN variants achieve strong performance, requiring the lowest memory consumptions comparable to SGD. Across all models, SWAN-0 surpasses the performance of the Adam and Momentum-`GradWhitening` (muon-like) baseline in terms of validation perplexities. SWAN† further delivers a comparable performance to Apollo series; and finally, SWAN‡ with NSDS scheme delivers the strongest, SOTA-level performance under this setup. Notably, on the 350M and 1.3B models, all SWAN variants reaches at least 2× speedup (in steps or tokens) relative to Adam (Figure 1 and Figure 9 (a)). Finally, we observe that in general, SWAN‡ has a slower convergence speed in the early stage than SWAN†. However, it offers strong tail convergence in return and surpasses SWAN† in terms of final performance.

## 5.2. Ablation of SWAN on LLM pretraining

We take SWAN-0 from the SWAN series as an example and conduct the following ablation studies.

**Effect of `GradNorm` and `GradWhitening` on Performance**   We consider six ablation settings: (1) SWAN-0 (2) SWAN-0 (`GradNorm` only), (3) SWAN-0 (`GradWhitening` only), (4) Adam, (5) Adam (momentum only), and (6) Adam (second moment only). As shown in Figure 4 (a), both `GradNorm` and `GradWhitening` contribute to SWAN's final performance. Removing either results in performance degradation. Similarly, Adam also requires first and second moment estimates for optimal performance.

**Does SWAN work only because it increases effective learning rate via `GradNorm`?**   To rule out this possibility, we remove the `GradNorm` operator from SWAN-0 and run a learning rate sweep. Starting with the default lr of SWAN-0, we apply multipliers from 1 to $10^3$. In Figure 4 (b), the blue line shows the final PPL for SWAN w/o `GradNorm` at different learning rates multipliers, while the dashed red line represents the performance of full SWAN-0 at the default learning rate. Although raising learning rate can improve the performance of SWAN-0 without `GradNorm`, the gap to full SWAN-0 remains large. This indicates that `GradNorm`'s gradient noise stabilization is essential and cannot be replaced simply by increasing the learning rate.

**How does warm-up affect the performance?**   Section 5.1 shows that SWAN can train with no warm-up phase even under a relatively large learning rate (0.001). Here, we compare Adam and SWAN with and without warm-ups. As seen in Figure 4 (c), SWAN without the warm-up phase gives better performance, and it still outperforms Adam under Adam's own warm-up schedule. On the other hand, Adam's performance decreases drastically without a proper warm-up. These suggest that SWAN is more robust to warm-

Table 2: Comparison with Adam and its memory-efficient low-rank variants on pre-training various sizes of LLaMA models on C4 dataset. Validation perplexity is reported, along with a memory estimate of the total of parameters and optimizer states based on BF16 format. Baseline results are directly taken from the official numbers reported in Zhao et al. (2024a); Zhu et al. (2024), as they shares exactly the same setup. Note that for Adam we report both the official results from Zhao et al. (2024a) and our reproduced result.

|  | 60M | 130M | 350M | 1.3 B |
|---|---|---|---|---|
| Adam | 33.02 (0.32G) | 24.44 (0.75G) | 19.24 (2.05G) | 16.44 (7.48G) |
| Adam (cited) | 34.06 (0.32G) | 25.08 (0.75G) | 18.80 (2.05G) | 15.56 (7.48G) |
| SWAN‡ | 30.59 (0.23G) | **22.61 (0.43G)** | **16.63 (0.93G)** | **13.56 (2.98G)** |
| SWAN† | **30.00 (0.23G)** | 22.83 (0.43G) | 17.14 (0.93G) | 14.42 (2.98G) |
| SWAN-0 | 32.28 (0.23G) | 24.13 (0.43G) | 18.22 (0.93G) | 15.13 (2.98G) |
| Momentum+`GradWhitening` | 31.6 (0.27G) | 24.59 (0.59G) | 19.30 (1.49G) | 16.08 (5.23G) |
| Galore | 34.88 (0.26G) | 25.36 (0.57G) | 18.95 (1.29G) | 15.64 (4.43G) |
| Apollo-mini | 31.93 (0.23G) | 23.53 (0.43G) | 17.18 (0.93G) | 14.17 (2.98G) |
| Apollo | 31.55 (0.26G) | 22.94 (0.57G) | 16.85 (1.29G) | 14.20 (4.43G) |
| Low-Rank | 78.18 (0.26G) | 45.51 (0.57G) | 37.41 (1.29G) | 142.53 (4.43G) |
| LoRA | 34.99 (0.36G) | 33.92 (0.80G) | 25.58 (1.76G) | 19.21 (6.17G) |
| ReLoRA | 37.04 (0.36G) | 29.37 (0.80G) | 29.08 (1.76G) | 18.33 (6.17G) |
| SWAN‡ speed up vs Adam | 1.52 X | 1.6 X | > 2.3 X | > 2.4 X |
| $r$ of low-rank methods | 128 | 256 | 256 | 512 |
| Training Steps | 10K | 20K | 60K | 100K |

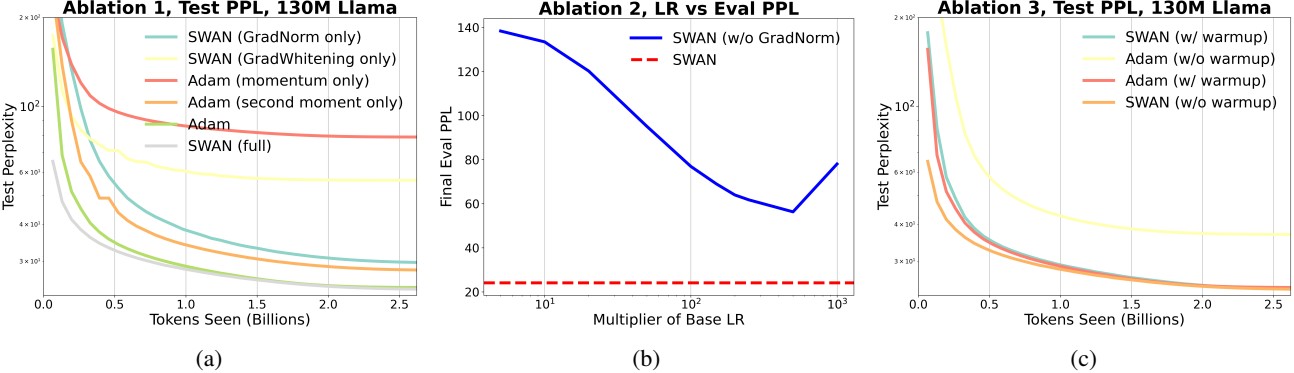

(a)            (b)            (c)

Figure 4: Ablation studies on 130M model. (a) On the contribution of each components in SWAN and Adam. (b) On removing `GradNorm` and compensate with larger learning rates. (c) On the effect of learning rate warm-ups.

up schedule and can train effectively with or without it.

**Is the speed-up Multiplicative or Additive?** A key question in assessing speedup factors is whether the improvement over Adam is *multiplicative* or *additive*. A multiplicative speedup implies that the optimizer's relative advantage remains proportionally consistent over time, while an additive speedup suggests a less desired constant step advantage. In Appendix J.2 and Figure 9, we show that for smaller models (60M and 130M), the empirical speedup trajectories indicate a primarily additive speedup. For larger models (350M and 1.3B), they suggest a multiplicative speedup.

### 5.3. Memory Efficiency and Throughput Analysis

**Memory Footprint** We compare SWAN-0, Adam, and Galore on a single A100 GPU. Unlike (Zhao et al., 2024a; Zhu et al., 2024), which report layer-wise training memory usage, we measure total end-to-end memory consumption under full-model training using a batch-size of 1 for LLaMA with 1.3B, 7B, and 13B parameters. As shown in Figure 1 (c), SWAN's memory usage is on par with SGD, providing nearly a 50% reduction in total memory. This underlines the benefit of the stateless design.

**Effective Throughput** We assess throughput when training a 1.3B LLama model on 8 A100 GPUs with a batch size of 130K. All gradient processing are done in BF16. We

Table 3: Raw and effective throughput analysis, under different model parallelization (MP) settings.

| Method | Raw / eff. throughput (1B) |
|---|---|
| Adam | 117872 / 117872 (tokens/s) |
| SWAN[†] w/o MP | 58600 / 117200 (tokens/s) |
| SWAN[‡] w/o MP | **107808 / 258739** (tokens/s) |
| SWAN[†] w/ MP | 114160 / 228320 (tokens/s) |
| SWAN[‡] w/ MP | **115872 / 278092** (tokens/s) |

use two metrics: *raw throughput*: number of tokens processed per second. *Effective throughput*: raw throughput adjusted by SWAN's token efficiency relative to Adam. These metrics evaluate the impact of different `GradWhitening` schemes on training speed, and also account for the fact that some optimizers make more effective use of training tokens. As shown in Table 3, SWAN[†] with naive NS `GradWhitening` requires model parallelization (where NS of different tensors are distributed to different GPUs) to achieve competitive throughput. Without model parallelization, the throughput of SWAN[†] is 50 % lower than Adam. With the proposed NSDS scheme, SWAN[‡] achieves an raw throughput comparable to Adam, *without* any need for model parallelization. Consequently, SWAN[‡] exhibits a $2 \times$ higher effective throughput than Adam.

### 5.4. Verifying the Theoretical insights of Section 4

We empirically validate the theoretical benefits of `GradNorm` and `GradWhitening`. As detailed in Appendix J.1, we demonstrate that `GradNorm` effectively stabilizes stochastic gradient distributions during SGD training of a scaled-down LLaMA model on the C4 dataset, evidenced by significantly reduced KL divergence fluctuations compared to standard training. Additionally, `GradWhitening` significantly enhances optimization performance across high-dimensional and ill-conditioned classic optimization functions, consistently outperforming both SGD and Adam. These findings confirm that `GradNorm` promotes gradient stability and `GradWhitening` addresses local curvature challenges, enabling faster and more reliable performance.

## 6. Beyond SWAN: the General Principle of Constructing Chain of Gradient Processors

Section 4 implied our thought process when designing SWAN: the operators `GradNorm` and `GradWhitening` were chosen to exploit the structures of transformer dynamics. However, the idea of preprocessing gradients via a chain of matrix operators is more fundamental. This raises a key question: given an arbitrary model, what gradient operators can we choose, so that the resulting optimizer is valid? After our ICML submission, the follow-up work of (Scetbon et al., 2025) introducing a general framework called multi-normalized gradient descent (MNGD). MNGD considers a set of $K \geq 1$ norm functions $(g_1, \ldots, g_K)$ and defines the update direction $\Delta W$ as the solution to:

$$- \arg\max_z \langle \boldsymbol{G}, z \rangle \text{ s.t. } \forall\, i \in [|1, K|],\ g_i(z) = 1 \,. \quad (8)$$

where $\boldsymbol{G}$ is the current gradient (can be raw gradient or momentum). In other words, MNGD chooses an update direction that is simultaneously normalized under all $K$ norms (each $g_i(z)$ is set to 1). (Scetbon et al., 2025) showed that solving this problem leads to an *iterative normalization* strategy one can loop over $i = 1$ to $K$ for several passes, normalizing the gradient with respect to each $g_i$ in turn.

Notably, SWAN fits into this framework as a special case. It can be viewed as a single-iteration approximation of MNGD with $K = 2$ norms: $g_1$ is the row-wise $\ell_2$ norm and $g_2$ is the spectral norm. This generalized viewpoint offers greater flexibility for designing new optimizers. As demonstrated in (Scetbon et al., 2025), one may design more elegant and efficient optimizers than SWAN (and other Shampoo variants). For example, instead of using a single spectral norm that requires expensive NS iterations, one could enforce multiple norms that have simpler projection operations.

## 7. Conclusion, Limitations and Future Work

We introduced SWAN, a stateless optimizer for LLM training that preprocesses raw gradients via a chain of two operators, `GradNorm` and `GradWhitening`. These design choices were intended to leverage structural redundancies in transformer training dynamics. Through theoretical analysis and empirical evidence, we demonstrated that SWAN maintains an SGD-level memory footprint while achieving performance on par with Adam. Notably, on the memory-efficient LLaMA training benchmark of Zhao et al. (2024a), when training 350M and 1B models, SWAN attained a $2\times$ speedup in tokens processed compared to Adam, marking a new state-of-the-art on this benchmark.

Our findings provide a first proof-of-concept that the stateless approach has the potential to serve as a practical, efficient alternative to optimizers that require tracking internal states. However, we acknowledge a gap between our setup and industry-scale LLM training (scale, context length, batch size, architecture, dataset, etc). These differences pose challenges for generalizing our insights, and we aim to address them in future work. Furthermore, future research may explore alternative design choices for stateless optimizers and extend SWAN to other complex training regimes beyond standard LLM pre-training.

## Impact Statement

This paper presents work whose goal is to advance the field of Machine Learning. There are many potential societal consequences of our work, none which we feel must be specifically highlighted here.

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

# A. Related works and extended discussions

**Towards Stateless Optimizers for LLM Training.** Adaptive optimizers generally rely on tracking internal state variables to perform weight updates, which can substantially increase memory consumption when training large models. Several recent works have successfully managed to reduce the memory requirements associated with storing additional state variables for training LLMs. Muon (Jordan et al., 2024; Bernstein & Newhouse, 2024b), a newly proposed optimizer, has demonstrated strong acceleration and memory saving for LLM training by simplifying shampoo-like optimizers (Gupta et al., 2018; Anil et al., 2021; Shi et al., 2023; Wang et al., 2024; Vyas et al., 2024; Peirson et al., 2022; Lin et al., 2024) and requiring only the tracking of a first-moment estimate. SGD-Sal (Xu et al., 2024b) only stores the first moment with a learning rate estimated at the beginning of training. Sign-based methods such as (Chen et al., 2023) have also demonstrated success on training transformer-based models by only tracking first moments. There are also several works that aim to enhance the memory efficiency of Adam by reducing the memory cost associated with second moments. Adam-mini (Zhang et al., 2024b) significantly reduces memory usage by storing only scalar values for each parameter block, while Adalayer (Zhao et al., 2024c) retains only the scalar average of the second moment for each layer. Alternatively, PSGD (Pooladzandi & Li, 2024; Li, 2017) focuses on exclusively tracking a pre-conditioner, eliminating the need to track a first moment estimate. Finally, the concurrent work of (Zhu et al., 2024) proposed an approximate gradient scaling method, that tracks the channel-wise or tensor-wise surrogate learning rates, further improving the memory efficiency. However, all the aforementioned optimizers still require the storage of state variables. In contrast, SWAN completely eliminates the need to store internal states for both the first and second moments by employing a combination of `GradNorm` and `GradWhitening` steps, which is discussed next.

**Pre-processing Gradients.** Gradient pre-processing is a common technique used to enhance performance of optimizers. Various pre-processing procedures have been proposed in the literature, such as signed gradient (Bernstein et al., 2018; Crawshaw et al., 2022; Chen et al., 2023; Kunstner et al., 2023), gradient clipping (Zhang et al., 2020), normalization (Zhang et al., 2020; You et al., 2019), and whitening (Yang & Laaksonen, 2008; Kingma & Ba, 2015; Hwang, 2024; Jordan et al., 2024; Bernstein & Newhouse, 2024c;a; Carlson et al., 2015). In this work, we particularly focus on normalization and whitening. We apply normalization row-wise on gradient matrices, similar to (You et al., 2019); together with gradient whitening under a specific structural assumption of the Fisher Information (Hwang, 2024; Martens et al., 2018), recovering the orthogonalization step used in (Jordan et al., 2024; Tuddenham et al., 2022). Our key result is that *composing* normalization and whitening on stochastic gradients is sufficient to enable the efficient training of LLMs in a completely stateless manner. Compared to Lamb (You et al., 2019), our normalization operation is applied on raw gradients instead of Adam states. Compared to Muon (Jordan et al., 2024), SWAN removes first-moment tracking and instead uses normalization, making the optimizer fully statless. Moreover, one of our proposed efficient NS scheme for computing square root inverse requires $\mathcal{O}(m^2)$ (assuming $m < n$) instead of $\mathcal{O}(m^3)$ of the standard Newton-Schulz scheme. We show empirically that removing any one of these two pre-processing steps from SWAN results in significant performance degradation (Section 5.2).

**The Role of Momentum** There are a growing body of work that showed the benefits of applying momentum for optimization (Jelassi & Li, 2022; Cutkosky & Mehta, 2020; Fu et al., 2023; Crawshaw et al., 2022; Zhang et al., 2020; Zhao et al., 2024d). While our work aims at reducing all optimizer states including momentum, we emphasize that our results does not contradict the effectiveness of momentum; in fact we found that enabling momentum with the suitable hyperparameter can indeed further improve the result. However, our core contribution is to push the boundaries and demonstrate that: it is possible to train LLMs matching the performance of Adam using a completely stateless optimizer. Whether enabling additional states for SWAN can further improve the performanceis orthogonal to our contribution. This provides more options under certain practical scenarios. For example, due to its memory efficiency, the stateless design of SWAN without momentum may allow training with 2X larger batch sizes compared with using momentum; depending on the users's application, the benefits of being able to run 2X larger batch size might out-weigh the benefits of momentum.

**Comparisons to Apollo optimizer (Zhu et al., 2024)** . A closely related concurrent work is Apollo (Zhu et al., 2024), which proposes to re-scales the raw gradient of SGD to achieve SGD-like memory. Given a 2D gradient matrix of shape $m$ by $n$, Apollo propose to perform structured $n-$ channel scaling (i.e., column-wise scaling), where the scaling factor is estimated via heuristic approaches similar to Fira (Chen et al., 2024). Compared to Apollo, SWAN enjoys the following advantages. 1), using the perspective of Apollo, SWAN can also be viewed as gradient rescaling $\mathbf{G} \cdot \frac{\texttt{GradWhitening}(\texttt{GradNorm}(\mathbf{G}))}{\mathbf{G}}$. The key difference here is, Apollo can only perform channel $n$ rescaling, and requires $\mathcal{O}(r(n+m))$ memory to maintain low-rank optimizer states to achieve channel $n$ rescaling. Apollo-mini further reduces the memory cost to $\mathcal{O}(n)$; however it can only

cope with channel-1 (global) scaling. On the contrary, the SWAN rescaling $\mathbf{G} \cdot \frac{\texttt{GradWhitening(GradNorm(G))}}{\mathbf{G}}$ achieves channel $m \cdot n$ rescaling with zero memory cost, offers richer representation power with much less memory cost. 2), In SWAN, the gradient preprocessing operators are chosen based on key theoretical insights of LLM learning dyanmics as discussed in Section 4; while the Apollo design is mainly based on heuristics. 3), in practice, SWAN has stronger performance in terms of LLM pretraining. In the same setup, SWAN‡ achieves 1.46X and 1.36X speedup versus Apollo-mini on 350M and 1.3B Llama models, respectively.

**Comparison to Muon**    Apart from algorithmic details, the key differences between our work and Muon (Bernstein & Newhouse, 2024b) is twofold. First, compared with Muon (Jordan et al., 2024), our core contribution is to push the boundaries and demonstrate that: it is possible to train LLMs matching the performance of Adam using a completely stateless optimizer. Whether our method can consistently outperform all existing non-Adam optimizers (such as shampoo/Muon/SOAP) is orthogonal to this contribution. Our work thus highlights a viable extreme stateless alternative for memory-constrained settings, and this distinction is central to our contribution. Second, the chain of thought behind the design choice of SWAN (exploiting structural redundancies of LLM dynamics to design composition of gradient processors) leads to fundamentally different research paths. From our view, the follow up work of (Gong et al., 2025) shows that the whitening operation used in both Muon and SWAN is a direct result of structured Fisher information matrix (FIM) approximation. As demonstrated in (Gong et al., 2025), this leads to potentially richer design choices that cannot be easily motivated within the normed steepest descent theory of Muon. Interestingly, (Gong et al., 2025) also shows that the structured FIM view leads to one additional coefficient to Muon ($\sqrt{\max(m, n)}$ rescaling) that were recently re-discovered in (Liu et al., 2025) (for both easier hyperparameter transfer from Adam and stabilizing training). Finally, SWAN can be seen as a fixed point solution of a iterative projection scheme (Row-normalization $\rightarrow$ whitening $\rightarrow$ row-normalization $\rightarrow$ whitening $\rightarrow$ ... ). This leads to a new theoretical framework that directly generalizes (Bernstein & Newhouse, 2024b). This is demonstrated in the other follow up work of (Scetbon et al., 2025), in which the authors shows that the SWAN-motivated formulation opens up more design spaces for optimizers. Specifically, (Scetbon et al., 2025) demonstrated that more elegant and efficient matrix-based optimizers can be easily derived. This is also briefly discussed in Section 6.

**Iterative methods for efficient whitening**    Computing `GradWhitening` exactly can be expensive, as it involves solving the matrix square-root inverse. In the literature, iterative solvers for inverse square root has been explored. One of the most used methods is the Newton-Schulz method (Song et al., 2022; Li et al., 2018; Huang et al., 2019), which is matrix-multiplication rich and allows a more GPU-friendly estimation. In the context of optimization, the same N-S procedure has been proposed in (Mei et al., 2023) and (Jackson, 2023) for preconditioned optimizer. An alternative N-S procedure has also been re-discovered in (Bernstein & Newhouse, 2024b), and optimized in (Jordan et al., 2024). Although our inital implementation was inspired by (Song et al., 2022; Li et al., 2018; Huang et al., 2019), in principle any estimators can also be used. Furthermore, we proposed a efficient heuristic scheme, focusing on condition number minimization as a proxy for whitening. The proposed scheme requires $\mathcal{O}(m^2)$ to estimate square root inverse instead of $\mathcal{O}(m^3)$ of the original N-S scheme, and offers competitive performance in the context of gradient preprocessing.

**Low-rank methods.**    Low-rank optimization techniques have been explored in the context of large language model (LLM) training as a means to reduce memory consumption. These methods focus on applying low-rank approximations to model weights, gradients, and/or optimizer state variables. A seminal work in this domain is LORA (Hu et al., 2021) to fine-tune pre-trained models using additional low-rank weight matrices at each layer, thereby significantly reducing memory usage to update the weights. More recently, methods such as ReLoRA (Lialin et al., 2023), FLORA (Hao et al., 2024), and Galore (Zhao et al., 2024a) have advanced low-rank optimization techniques for memory-efficient LLM pre-training. These approaches leverage low-rank gradient projections to enable full-rank learning, thereby achieving memory savings without compromising model capacity. Notably, they have achieved substantial reductions in the memory consumption of optimizer states, with only minimal impact on model performance. While these approaches effectively reduce the memory footprint of LLM training, they still necessitate storing internal states, resulting in higher memory consumption compared to SWAN.

## B. Desired properties of adaptive optimizers

There is a rich literature on understanding adaptive methods' inner workings and unreasonable effectiveness. Using Adam as an example, we first summarize from the literature below the key desired properties of stateful adaptive optimizers that contribute to their empirical success: *gradient smoothing*, *gradient invariance*, and *gradient whitening*. Then we discuss how these understandings will leads to the design of *stateless* adaptive optimizers.

**Gradient Smoothing.** Under the stochastic optimization setting, mini-batch sampling introduces heterogeneous distribution shift on the gradient distribution: $\mathbf{G}^{(t)} = \mathbb{E}[\mathbf{G}^{(t)}] + \boldsymbol{\varepsilon}^{(t)}$, where $\boldsymbol{\varepsilon}^{(t)}$ is time-heterogeneous noise induced by mini-batch sampling. While $\boldsymbol{\varepsilon}^{(t)}$ helps SGD escapes local optima (Jastrzębski et al., 2017; Zhu et al., 2018), the *covariate shift* of $\boldsymbol{\varepsilon}^{(t)}$ over time also present challenges to learning as the model needs to adjust and compensate for this shift, especially under the emergence of heavy tailed gradient distributions (Zhang et al., 2020) [7]. Following this viewpoint, it has been proven that momentum reduces the influence of noises for SGD (Cutkosky & Mehta, 2020; Crawshaw et al., 2022). Therefore we hypothesis that the first moment estimate $\boldsymbol{m}^{(t)}$ of Adam also effectively stabilizes gradient distribution and reduces effect of $\boldsymbol{\varepsilon}^{(t)}$. This smoothing stabilizes the variance caused by noisy stochastic gradients across time.

**Gradient Invariance.** More recently it has also been identified (Kunstner et al., 2023; 2024) that the major factor contributing to the performance gap between SGD and Adam might lie in Adam's *Sign-descent*-like nature (Bernstein et al., 2018; Crawshaw et al., 2022; Chen et al., 2023). Intuitively, Adam without bias correction under $\beta_1 = 0$ and $\beta_2 = 0$ is equivalent to signed gradient descent ($\Delta\mathbf{W} = \text{sign}(\mathbf{G})$). Indeed, the performance of Adam can be closely reproduced (Kunstner et al., 2023; Crawshaw et al., 2022) or even surpassed (Chen et al., 2023) by variants of signed descent with momentum. Apart from sign-based methods, evidence on performance boost using gradient clipping/normalization was also discussed in the context of understanding Adam (Zhang et al., 2020). Therefore, we hypothesize that one of the key properties of Adam is that it offers *invariance over certain transformations* on gradients. Particularly, the original Adam is invariant to diagonal rescaling of the gradients (Kingma & Ba, 2015); the signed gradient method is invariant to *any* scaling that preserves the sign of gradients; and the clipped SGD variant is invariant to extreme gradient magnitude spikes.

**Gradient Whitening.** Finally, we argue that the empirical success of adaptive methods also lies in that they model the curvature by first-order information. This is realized by the second moment estimate $\boldsymbol{\nu}^{(t)}$, which approximates the diagonal of the Fisher information matrix (Kingma & Ba, 2015; Hwang, 2024); helping to counteract local curvatures of the problem. Specifically, Adam computes a trailing estimation of the diagonal coefficients of the Fisher matrix $\mathbf{F} = \mathbb{E}[\boldsymbol{g}\boldsymbol{g}^\top]$ by tracking $\hat{\mathbf{F}} = \text{diag}(\mathbf{F}) = \text{diag}[\mathbb{E}[\boldsymbol{g}^2]]$, where $\boldsymbol{g} = \text{vec}(\mathbf{G})$ is the vectorized gradient. Interestingly, instead of preconditioning the first moment as $\hat{\mathbf{F}}^{-1}\text{vec}(\boldsymbol{m})$, Adam uses a whitening-like preconditioned update $\hat{\mathbf{F}}^{-\frac{1}{2}}\text{vec}(\boldsymbol{m})$, suggesting an *element-wise* approximate whitening of the gradient. It has been shown that such element-wise whitening leads to diagonal approximation to inverse Hessian $\hat{\mathbf{F}}^{-\frac{1}{2}} \approx \text{diag}(\mathbf{H}^{-1})$ (Molybog et al., 2023). Recent empirical studies show that Adam biases optimization trajectories towards regions where the condition number of Hessian is low (Jiang et al., 2024). Therefore, we hypothesize that Adam approximately whitens the gradients element-wise, leading to well-conditioned regions.

## C. Additional Analysis

### C.1. Analyzing the GradWhitening Pt. II: Robustness Against Local Curvature

In this section, we present main results regarding the convergence rate of the `GradWhitening` method, understand its implications, and compare it with the lower bounds of GD and Adam.

First, for simplicity, we focus on the following quadratic problem:

$$\mathcal{L}(\mathbf{W}) = \frac{1}{2}\text{Tr}(\mathbf{W}^\top\mathbf{H}\mathbf{W}) - \text{Tr}(\mathbf{C}^\top\mathbf{W}), \tag{9}$$

where $\mathbf{W} \in \mathbb{R}^{m \times n}$ is the parameter matrix, $\mathbf{H} \in \mathbb{R}^{m \times m}$ is a positive definite matrix, and $\mathbf{C} \in \mathbb{R}^{m \times n}$ is a constant matrix.

For simplicity and without loss of generality, we assume $\mathbf{C} = 0$. This is because minimizing $\mathcal{L}(\mathbf{W}) = \frac{1}{2}\text{Tr}(\mathbf{W}^\top\mathbf{H}\mathbf{W}) - \text{Tr}(\mathbf{C}^\top\mathbf{W})$ is equivalent to minimizing $\mathcal{L}(\mathbf{W}) = \frac{1}{2}\text{Tr}[(\mathbf{W} - \mathbf{W}^*)^\top\mathbf{H}(\mathbf{W} - \mathbf{W}^*)]$, where $\mathbf{W}^* = \mathbf{H}^{-1}\mathbf{C}$. By defining $\mathbf{Z} = \mathbf{W} - \mathbf{W}^*$, the problem reduces to minimizing $\mathcal{L}(\mathbf{Z}) = \frac{1}{2}\text{Tr}(\mathbf{Z}^\top\mathbf{H}\mathbf{Z})$.

**Remark** Most results in this note can be easily extended to any loss function that are either i) strongly convex; or ii) has twice differentiable functions and Lipschitz continuous Hessian, by considering their the second order approximation around $\mathbf{W}^*$.

Next, to understand the effect of `GradWhitening`, we will examine the gradient flow dynamics induced by `GradWhitening`.

---

[7]Such shift cannot be removed by forward covariate-shift reduction architectures such Layer Norm, as it is only invariant to global scaling and re-centering, such as $\mathbf{W}^{(t)} = \delta\mathbf{W}^{(t)} + \boldsymbol{\gamma}\mathbf{1}^\top$ for some scalar $\delta$ and incoming vector shift $\boldsymbol{\gamma}$ (Ba et al., 2016).

Consider the `GradWhitening`-modified gradient descent:

$$\Delta\mathbf{W}^{(t)} = -\eta\texttt{GradWhitening}(\mathbf{G}^{(t)}) \tag{10}$$

its exact convergence rate is given by the result as below:

**Theorem 2** (**Contraction factor of** `GradWhitening`). *Consider the quadratic loss function Equation* (9). *Assume the initialization distribution of* $\mathbf{W}^0$ *assigns zero probability to any set of zero Lebesgue measure in* $\mathbb{R}^{m\times n}$. *Let our update rule be:*

$$\mathbf{W}_{whitened}^{(t+1)} = \mathbf{W}_{whitened}^{(t)} - \eta\texttt{GradWhitening}(\mathbf{G}^{(t)})$$

*where the learning rate is* $\eta$. *Then, with probability 1, we have:*

- *The optimal dynamic learning rate to achieve the fastest convergence is given by*

$$\eta^{(t)^*} = \frac{\|\mathbf{H}\mathbf{W}_{whitened}^{(t)}\|_1}{Tr[\mathbf{H}]}. \tag{11}$$

  *where* $\|\mathbf{H}\mathbf{W}_{whitened}^{(t)}\|_1$ *denotes the Schatten p-norm with* $p = 1$ *(i.e., sum of singular values).*

- *Under* $\eta^{(t)^*}$, *the contraction factor of loss function at* $t$ *is given by:*

$$\frac{\mathcal{L}(\mathbf{W}_{whitened}^{(t+1)}) - \mathcal{L}^*}{\mathcal{L}(\mathbf{W}_{whitened}^{(t)}) - \mathcal{L}^*} = 1 - \frac{\|\mathbf{H}\mathbf{W}_{whitened}^{(t)}\|_1^2}{Tr[(\mathbf{W}_{whitened}^{(t)})^\top\mathbf{H}\mathbf{W}_{whitened}^{(t)}]Tr[\mathbf{H}]} \tag{12}$$

- *Furthermore, if we additionally enforce* $\mathbf{W}^0 \sim V^{m\times n}(\mathbb{R})$, *i.e., initialized as an element in Steifel manifold. Then we have*

$$\frac{\mathcal{L}(\mathbf{W}_{whitened}^{t=1}) - \mathcal{L}^*}{\mathcal{L}(\mathbf{W}^0) - \mathcal{L}^*} = 0 \tag{13}$$

*That is,* `GradWhitening` *solves the optimization problem (Equation* (9)*) with 1 step iteration.*

Theorem 2 has the following key implications.

**Convergence rate is condition number agnositc** Unlike the convergence rates of GD and Adam presented in Zhang et al. (2024a), as well as Theorem 3 and Corollary 1 in Appendix, the optimal convergence rate (12) of `GradWhitening` no longer explicitly depends on the condition number $\kappa$ of $H$. In fact, consider a lower bound $\frac{\|\mathbf{H}\mathbf{W}_{\text{whitened}}^{(t)}\|_1^2}{Tr[(\mathbf{W}_{\text{whitened}}^{(t)})^\top\mathbf{H}\mathbf{W}_{\text{whitened}}^{(t)}]Tr[\mathbf{H}]} \geq \frac{Tr[H\mathbf{W}_{\text{whitened}}^{(t)}]^2}{Tr[(\mathbf{W}_{\text{whitened}}^{(t)})^\top\mathbf{H}\mathbf{W}_{\text{whitened}}^{(t)}]Tr[\mathbf{H}]}$, since trace of $H$ appear both in the nominator and denominator, we expect that to be more robust to ill-conditioned problems. For example, consider the specific initialization $\mathbf{W}_{\text{whitened}}^{(t)} = cI$, it is straightforward to show that $\frac{\|\mathbf{H}\mathbf{W}_{\text{whitened}}^{(t)}\|_1^2}{Tr[(\mathbf{W}_{\text{whitened}}^{(t)})^\top\mathbf{H}\mathbf{W}_{\text{whitened}}^{(t)}]Tr[\mathbf{H}]} \geq \frac{Tr[H\mathbf{W}_{\text{whitened}}^{(t)}]^2}{Tr[(\mathbf{W}_{\text{whitened}}^{(t)})^\top\mathbf{H}\mathbf{W}_{\text{whitened}}^{(t)}]Tr[\mathbf{H}]} \perp \kappa$, which is completely disentangled from the condition number. Hence $\frac{\|\mathbf{H}\mathbf{W}_{\text{whitened}}^{(t)}\|_1^2}{Tr[(\mathbf{W}_{\text{whitened}}^{(t)})^\top\mathbf{H}\mathbf{W}_{\text{whitened}}^{(t)}]Tr[\mathbf{H}]}$ would not shrink as $\kappa \to \infty$. See Proposition 2 for less extreme situations.

**Superlinear convergence with Stiefel manifold initialization** Theorem 2 suggests that if $\mathbf{W}_{\text{whitened}}^{(t)}$ is initialized in the Stiefel manifold, then `GradWhitening` reaches superlinear convergence rate (= Newton's method), while being cheaper. In fact, it is straightforward to verify that `GradWhitening` reaches optimal solution with 1 step update. This implies `GradWhitening` is theoretically the optimal optimization algorithm if $\mathbf{W}$ is initialized in the Stiefel manifold.

**Estimation and interpretation of optimal learning rate** Compared to the optimal dynamic learning rate of gradient descent $G = \frac{G^\top G}{G^\top HG}$, the optimal learning rate $\eta^{(t)^*}$ of `GradWhitening` is much easier to compute. $\frac{Tr[H\mathbf{W}_{\text{whitened}}^{(t)}]}{Tr[\mathbf{H}]}$ can be seen as balancing the average gradient magnitude against the average curvature. A higher trace of gradient ($\mathbf{H}\mathbf{W}_{\text{whitened}}^{(t)}$) (strong gradients) relative to $\mathbf{H}$ (steep curvature) suggests a larger learning rate, promoting faster updates. Conversely, a higher trace of $\mathbf{H}$ would imply a smaller learning rate to ensure stable convergence in regions with high curvature.

Next, we show that the convergence speed of `GradWhitening` update is indeed robust to the condition number of local curvature.

**Proposition 2** (**Robustness of** `GradWhitening` **update convergence rate against the condition number of local Hessian**). *Consider the quantity:*

$$Q := \frac{Tr[\mathbf{H}\mathbf{W}^t_{whitened}]^2}{Tr[(\mathbf{W}^{(t)}_{whitened})^T\mathbf{H}\mathbf{W}^{(t)}_{whitened}]Tr[\mathbf{H}]}$$

*Assume: i)* ,$\mathbf{W}^{(t)}_{whitened} \neq \mathbf{W}^*$*; and ii) the norm of* $\mathbf{H}$ *is bounded. Then, there exist some finite positive constant c, such that*

$$Q > c$$

*This holds even if* $\kappa \to +\infty$*, where* $\kappa$ *is the condition number of* $\mathbf{H}$*.*

**Remark** Proposition 2 does not that the bound cannot be arbitrarily small, but rather: a small condition number does not necessarily lead to a diminished bound. This distinguishes our method from standard SGD, where such a decrease would be expected.

Below, we provide comparison between `GradWhitening` modified gradient descent and Adam. We only consider non-Stiefel initialization for `GradWhitening`, since with non-Stiefel initialization `GradWhitening` is optimal according to Theorem 2. Our results below shows that, for poor conditioned problems `GradWhitening` with a properly chosen single global learning rate always outperforms Adam even with *optimally tuned sub-group learning rates*, in terms of convergence speed.

**Proposition 3** (`GradWhitening` **with single lr vs Adam with tuned group lr**). *Consider the optimization problem Equation* (9)*. Assume* $\mathbf{H}$ *is block-diagonal, i.e.,* $\mathbf{H} = diag(\mathbf{H}_1, \mathbf{H}_2, \ldots, \mathbf{H}_L)$*, where each* $\mathbf{H}_l \in \mathbb{R}^{m_l \times m_l}$ *is a positive definite matrix for* $l = 1, 2, \ldots, L$*, and* $\sum_{l=1}^{L} m_l = m$*. Assuming for* `GradWhitening` *we use one global learning rate for all parameters; and for Adam, we use the optimally chosen group learning rate* $\eta_l$ *and initial condition* $w_0$ *for each block* $\mathbf{H}_l$*.*

*Assume either if i) certain regularity conditions are met (see proof in Appendix), or ii), if* $\mathbf{H}$ *is poorly-conditioned (its condition number is large enough). Then: regardless of its initialization,* `GradWhitening` *with a properly chosen learning rate will still have a strictly better convergence speed (i.e., smaller contraction factor) across all blocks* $l \in [L]$ *than Adam* ($\beta_1 = 0, \beta_2 = 1$) *under optimal group-wise learning rates and initial condition.*

**Remark** As pointed out by (Zhang et al., 2024a) and (Da Silva & Gazeau, 2020), Adam with $\beta_2 < 1$ will have issues with convergence, which will not be completely removed even with lr decay. Therefore, we will not discuss the case of $\beta_2 < 1$ to avoid the complication.

### C.2. Numerical Verification of Proposition 1

Given a STB, we consider the following standard full-batch learning dynamics (Tian et al., 2023). Define the conditional expectation $\mathbb{E}_{q=m}[\cdot] := \mathbb{E}[\cdot|q = m]$. Consider the dynamics of the weight matrix $W$ and the attention logits $z_q$, if we train the model with a batch of inputs that always end up with query $q[i] = m$. The weight update for $W$ and $z_q$ are given by the following noisy updates:

$$\dot{\mathbf{W}}^{(t)} = \mathbb{E}_{q=m}\left[\boldsymbol{f}^{(t)}(\mathbf{G}_{\boldsymbol{h}} \odot \boldsymbol{h}'^{(t)})^\top\right], \quad \dot{\boldsymbol{z}}_m^{(t)} = \mathbb{E}_{q=m}\left[\left(\frac{\partial \boldsymbol{b}}{\partial \boldsymbol{z}_m^{(t)}}\right)^\top \mathbf{U}_C^\top \boldsymbol{g}_f^{(t)}\right], \tag{14}$$

Where $\boldsymbol{f}^{(t)} = \left(\mathbf{U}_C\left(\exp(\boldsymbol{z}_q^{(t)}) \odot \boldsymbol{x}\right) + \boldsymbol{u}_q\right)$, $(\boldsymbol{h}^{(t)})' = \phi'((\mathbf{W}^{(t)})^\top\boldsymbol{f}^{(t)})$ is the derivative of the current activation, $\mathbf{G}_{\boldsymbol{h}}^{(t)} = \nabla_{\boldsymbol{h}^{(t)}}\mathcal{L}$ is the gradient of the loss function $\mathcal{L}$ with respect to the hidden activation $\boldsymbol{h}^{(t)}$, and $\boldsymbol{g}_{\boldsymbol{f}^{(t)}}^{(t)} = \sum_k \boldsymbol{g}_{\boldsymbol{h}_k^{(t)}}^{(t)}(\boldsymbol{h}_k^{(t)})'\boldsymbol{w}_k^{(t)}$ is the sum of the gradients with respect to the attention logits. Here, $\boldsymbol{w}_k^{(t)}$ is the $k$-th column of $\mathbf{W}^{(t)}$, $\boldsymbol{g}_{\boldsymbol{h}_k^{(t)}}^{(t)}[i]$ be the backpropagated gradient sent to node $k$ at sample $i$.

Then, we numerically solving the STB ODE with $n = 12, M_C = 10$ in Equation (14). During all training steps, we analytically track the evolution of Hessian of $rmW$. Results are shown in Figure 5. As predicted by Proposition 1, we see very similar structures across the diagonal blocks of the Hessian.

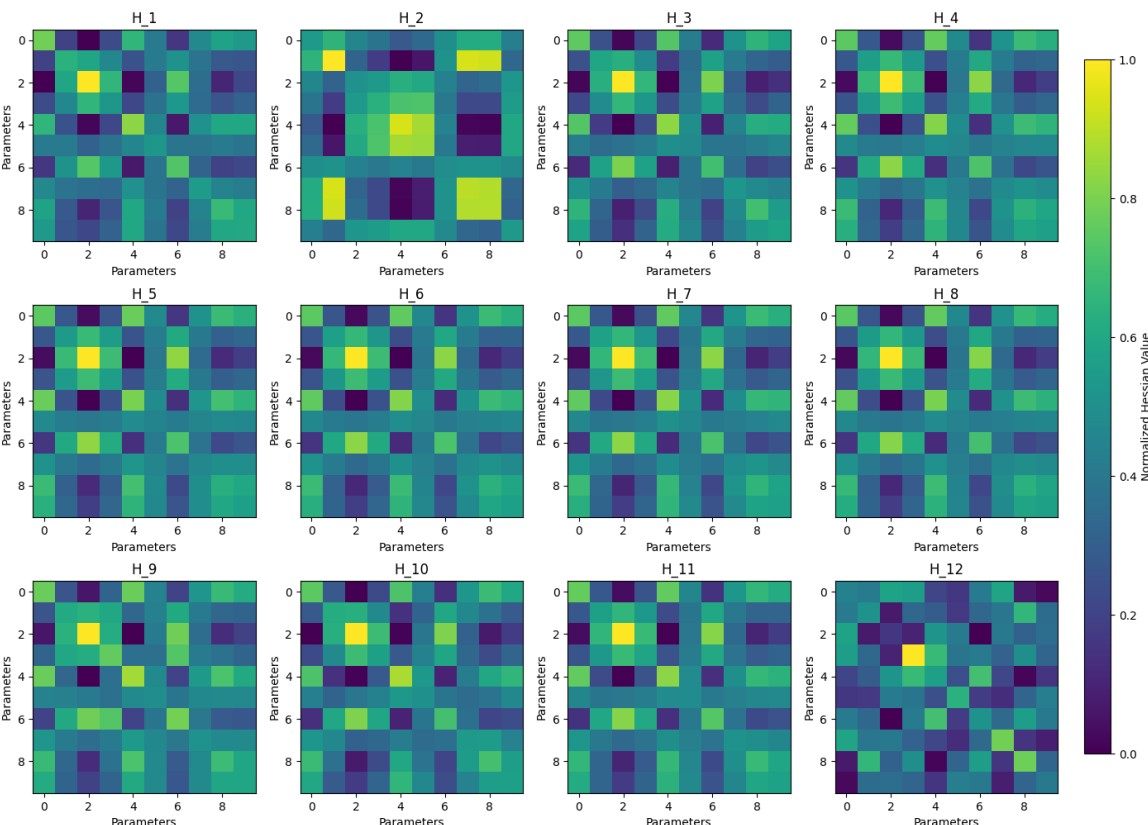

Figure 5: Normalized Hessian Blocks of size $M_C \times M_C$ along the diagonal direction of the Hessian, obtained from numerically solving the STB ODE (with $n = 12, M_C = 10$) (1) given by the full-batch dynamics (i.e., removing noise in Equation (14)). During all training steps, we analytically track the evolution of Hessian. As predicted by Proposition 1, we see very similar structures across the diagonal blocks of the Hessian.

## D. Proof of Theorem 1

*Proof.* We first consider the noiseless, full batch dynamics. Define $\mathbf{V} \in \mathbb{R}^{M_C \times n}$ as $\mathbf{V} := \mathbf{U}_C^\top \mathbf{W}$. Then following Theorem 2 in (Tian et al., 2023), each column of $\mathbf{V}$ satisfies the following differential equation:

$$\dot{\mathbf{V}}_{[:,j]} = \exp(\mathbf{V}_{[:,j]}^2/2 + C) \odot \mathbb{E}_q[g_{h_j}\boldsymbol{x}] \tag{15}$$

The corresponding dynamics of attention score is given by:

$$\boldsymbol{z}_q = \frac{1}{2}\sum_j \mathbf{V}_{[:,j]}^2. \tag{16}$$

Without loss of generality, in this proof we only consider $C = 0$.

Now, following the argument of Lemma B.6 of (Zhao et al., 2024a), we reparameterize the dynamics *row-wise*. For this, consider instead

$$\mathbf{V} = \begin{bmatrix} \boldsymbol{u}_1^\top \\ \boldsymbol{u}_2^\top \\ \vdots \\ \boldsymbol{u}_{M_C}^\top \end{bmatrix}$$

Then, equation 15 becomes:

$$\dot{\boldsymbol{u}}_i = [\exp(\boldsymbol{u}_i{}^2) \cdot \mathbf{1}]\boldsymbol{\mu}_i \tag{17}$$

where $\boldsymbol{\mu}_i \in \mathbb{R}^{n \times 1}$ is given by $[\boldsymbol{\mu}_i]_j := \mathbb{E}_q[g_{h_j} x_i]$. Therefore, it is clear that $\boldsymbol{u}_i$ always move along the direction of $\boldsymbol{\mu}_i$ due to the stationary back-propagated gradient assumption. Hence, $\dot{\boldsymbol{u}}_i = \alpha_i(t)\boldsymbol{\mu}_i$ for some scalar dynamics $\alpha_i(t)$.

Next, consider the mini-batch version of the dynamics. In this case, the packpropagated gradient term $[\boldsymbol{\mu}_i]_j := \mathbb{E}_q[g_{h_j} x_i]$ is corrupted by some i.i.d. mini-batch noise $\boldsymbol{\xi}$. The noisy row-wise dynamics now becomes:

$$\dot{\boldsymbol{u}}_i = \alpha_i(t)(\boldsymbol{\mu}_i + \boldsymbol{\xi}_i) \tag{18}$$

Therefore, after row-wise standardization, the new dynamics becomes

$$\dot{\tilde{\boldsymbol{u}}}_i = \frac{\alpha_i(t)(\mu_i + \boldsymbol{\xi}_i) - \alpha_i(t)(\frac{1}{n}\sum_j \mu_{ij} + \frac{1}{n}\sum_j \xi_{ij})}{\alpha_i(t)(\frac{1}{n}\sum_j (\mu_{ij} + \xi_{ij} - \frac{1}{n}\sum_j \mu_{ij} - \frac{1}{n}\sum_j \xi_{ij})^2)}$$

$$= \frac{(\mu_i + \boldsymbol{\xi}_i) - (\frac{1}{n}\sum_j \mu_{ij} + \frac{1}{n}\sum_j \xi_{ij})}{(\frac{1}{n}\sum_j (\mu_{ij} + \xi_{ij} - \frac{1}{n}\sum_j \mu_{ij} - \frac{1}{n}\sum_j \xi_{ij})^2)}$$

Therefore, the normalized noisy gradient $\dot{\tilde{\boldsymbol{u}}}_i$ no longer depend on the time-variant component $\alpha(t)$. Hence, we have proved:

$$\mathrm{Cov}[\tilde{\mathbf{G}}_{\mathbf{U}_C^\top \mathbf{W}}[i, :]^{(t_1)}] = \mathrm{Cov}[\tilde{\mathbf{G}}_{\mathbf{U}_C^\top \mathbf{W}}[i, :]^{(t_2)}] \quad \text{for all } t_1, t_2, \text{ and } i.$$

The corresponding result for $\tilde{\mathbf{G}}_{\boldsymbol{z}_q}^{(t)} := \mathtt{GradNorm}(\frac{\partial \mathcal{L}_{\mathbf{W}, \boldsymbol{z}_q}(\top \boldsymbol{x}^{(t)})}{\boldsymbol{z}_q})$ can be trivially derived due to Equation (16).

$\square$

# E. Proof of Theorem 2

*Proof.* We first show that $\nabla \mathcal{L}(\mathbf{W}^{(0)}) = \mathbf{H}\mathbf{W}^{(0)}$ (and hence $\nabla \mathcal{L}(\mathbf{W}_{\text{whitened}}^{(t)})$ with $t \neq \infty$) are non-zero with probability 1 under Assumption of the theorem. Given $\nabla \mathcal{L}(\mathbf{W}^{(0)}) = \mathbf{H}\mathbf{W}^{(0)}$, the set of matrices $\mathbf{W}^{(0)}$ such that $\mathrm{Tr}(\mathbf{H}\mathbf{W}^{(0)}) = 0$ forms a hyperplane in the space of $d \times d$ matrices. Specifically, it is defined by the linear equation: $\mathrm{Tr}(\mathbf{H}\mathbf{W}^{(0)}) = 0$. Since $\mathbf{H}$ is positive definite, at least one entry of $\mathbf{H}$ is non-zero. Thus, the hyperplane $\mathrm{Tr}(\mathbf{H}\mathbf{W}^{(0)}) = 0$ has zero Lebesgue measure in the space of $d \times d$ matrices. Given that $\mathbf{W}^{(0)}$ is sampled from a continuous distribution, the probability that $\mathrm{Tr}(\mathbf{H}\mathbf{W}^{(0)}) = 0$ is zero. Therefore, $\nabla \mathcal{L}(\mathbf{W}^{(0)}) \neq 0$ (and hence $\nabla \mathcal{L}(\mathbf{W}_{\text{whitened}}^{(t)})$ with $t \neq \infty$) with probability 1.

Next, we define the cost-to-go as:

$$\mathcal{L}(\mathbf{W}^{(t)}) - \mathcal{L}^* = \frac{1}{2}\mathrm{Tr}\left[(\mathbf{W}^{(t)})^\top \mathbf{H}\mathbf{W}^{(t)}\right],$$

and the per-step improvement is (since $\mathcal{L}^* = 0$ under $\mathbf{W} = 0$, ):

$$\mathcal{L}(\mathbf{W}^{(t)}) - \mathcal{L}(\mathbf{W}^{(t+1)}) = \frac{1}{2}\mathrm{Tr}\left[(\mathbf{W}^{(t)})^\top \mathbf{H}\mathbf{W}^{(t)}\right] - \frac{1}{2}\mathrm{Tr}\left[(\mathbf{W}^{(t+1)})^\top \mathbf{H}\mathbf{W}^{(t+1)}\right].$$

Substituting the update rule $\mathbf{W}^{(t+1)} = \mathbf{W}^{(t)} - \eta \mathtt{GradWhitening}(\mathbf{G}_{\text{whitened},l}) = \mathbf{W}^{(t)} - \eta\mathbf{U}\mathbf{V}^\top$, we get:

$$\mathcal{L}(\mathbf{W}^{(t)}) - \mathcal{L}(\mathbf{W}^{(t+1)}) = \frac{1}{2}\mathrm{Tr}\left[(\mathbf{W}^{(t)})^\top \mathbf{H}\mathbf{W}^{(t)}\right] - \frac{1}{2}\mathrm{Tr}\left[(\mathbf{W}^{(t)} - \eta\mathbf{U}\mathbf{V}^\top)^\top \mathbf{H}(\mathbf{W}^{(t)} - \eta\mathbf{U}\mathbf{V}^\top)\right].$$

Expanding the right-hand side, we have

$$\mathcal{L}(\mathbf{W}^{(t)}) - \mathcal{L}(\mathbf{W}^{(t+1)}) = \eta\mathrm{Tr}\left[(\mathbf{W}^{(t)})^\top \mathbf{H}\mathbf{U}\mathbf{V}^\top\right] - \frac{\eta^2}{2}\mathrm{Tr}\left[(\mathbf{U}\mathbf{V}^\top)^\top \mathbf{H}(\mathbf{U}\mathbf{V}^\top)\right].$$

Now, noticing that $\mathbf{G} = \mathbf{H}\mathbf{W}^{(t)} = \mathbf{U}\Sigma\mathbf{V}^\top$, we have:

$$\mathrm{Tr}\left[(\mathbf{W}^{(t)})^\top\mathbf{H}\mathbf{U}\mathbf{V}^\top\right] = \mathrm{Tr}\left[(\mathbf{H}\mathbf{W}^{(t)})^\top\mathbf{U}\mathbf{V}^\top\right] = \mathrm{Tr}\left[(\mathbf{U}\Sigma\mathbf{V}^\top)^\top\mathbf{U}\mathbf{V}^\top\right] = \mathrm{Tr}\left[\mathbf{V}\Sigma\mathbf{U}^\top\mathbf{U}\mathbf{V}^\top\right] = \mathrm{Tr}\left[\mathbf{V}\Sigma\mathbf{V}^\top\right].$$

Since $\mathbf{V}$ is orthogonal, $\mathbf{V}^\top\mathbf{V} = \mathbf{I}$, and $\Sigma$ is diagonal, we obtain:

$$\mathrm{Tr}\left[(\mathbf{W}^{(t)})^\top\mathbf{H}\mathbf{U}\mathbf{V}^\top\right] = \mathrm{Tr}(\Sigma) = \|\mathbf{H}\mathbf{W}^{(t)}_{\text{whitened}}\|_1.$$

Similarly:

$$\mathrm{Tr}\left[(\mathbf{U}\mathbf{V}^\top)^\top\mathbf{H}(\mathbf{U}\mathbf{V}^\top)\right] = \mathrm{Tr}\left[\mathbf{V}\mathbf{U}^\top\mathbf{H}\mathbf{U}\mathbf{V}^\top\right] = \mathrm{Tr}\left[\mathbf{V}\Lambda\mathbf{V}^\top\right] = \mathrm{Tr}(\mathbf{H}),$$

where $\Lambda$ is the eigenvalue matrix of $\mathbf{H}$. Given those intermediate results, we have:

$$\frac{\mathcal{L}(\mathbf{W}^{(t+1)}) - \mathcal{L}^*}{\mathcal{L}(\mathbf{W}^{(t)}) - \mathcal{L}^*} = 1 - \frac{\mathcal{L}(\mathbf{W}^{(t)}) - \mathcal{L}(\mathbf{W}^{(t+1)})}{\mathcal{L}(\mathbf{W}^{(t)}) - \mathcal{L}^*}$$

$$= 1 - \frac{\eta\|\mathbf{H}\mathbf{W}^{(t)}_{\text{whitened}}\|_1 - \frac{\eta^2}{2}\mathrm{Tr}(\mathbf{H})}{\frac{1}{2}\mathrm{Tr}\left[(\mathbf{W}^{(t)})^\top\mathbf{H}\mathbf{W}^{(t)}\right]}.$$

Noticing that this is a quadratic function of $\eta$ and the second order coefficient is positive, it is straightforward to verify via the quadratic formula that the optimal learning rate is given by

$$\eta_t^* = \frac{\|\mathbf{H}\mathbf{W}^{(t)}_{\text{whitened}}\|_1}{\mathrm{Tr}(\mathbf{H})}.$$

Under which the optimal contraction factor is given by

$$\frac{\mathcal{L}(\mathbf{W}^{(t+1)}_{\text{whitened}}) - \mathcal{L}^*}{\mathcal{L}(\mathbf{W}^{(t)}_{\text{whitened}}) - \mathcal{L}^*} = 1 - \frac{\|\mathbf{H}\mathbf{W}^{(t)}_{\text{whitened}}\|_1^2}{\mathrm{Tr}\left[(\mathbf{W}^{(t)}_{\text{whitened}})^\top\mathbf{H}\mathbf{W}^{(t)}_{\text{whitened}}\right]\mathrm{Tr}(\mathbf{H})}.$$

Finally, if we additionally enforce $\mathbf{W}^{(0)} \sim V^{m\times n}(\mathbb{R})$, i.e., we can parameterize $\mathbf{W}^{(0)} = \mathbf{O}$ where $\mathbf{O}$ is orthogonal, then it is trivial to verify that `GradWhitening` reaches the optimal solution with a 1-step update. To see this, consider the `GradWhitening` update:

$$\mathbf{W}^{(1)}_{\text{whitened}} = \mathbf{O} - \eta^*\texttt{GradWhitening}(\mathbf{H}\mathbf{O}) = \mathbf{O} - \frac{\|\mathbf{H}\mathbf{O}\|_1}{\mathrm{Tr}(\mathbf{H})}\texttt{GradWhitening}(\mathbf{H}\mathbf{O}),$$

noticing that $\frac{\|\mathbf{H}\mathbf{O}\|_1}{\mathrm{Tr}(\mathbf{H})} = 1$, and $\texttt{GradWhitening}(\mathbf{H}\mathbf{O}) = \mathcal{P}(\mathbf{Q}\Lambda\mathbf{Q}^\top\mathbf{O}) = \mathbf{Q}\mathbf{Q}^\top\mathbf{O} = \mathbf{O}$. Hence:

$$\mathbf{W}^{(1)}_{\text{whitened}} = \mathbf{O} - \eta^*\texttt{GradWhitening}(\mathbf{H}\mathbf{O}) = \mathbf{O} - \mathbf{O} = \mathbf{0} = \mathbf{W}^*.$$

Hence, the proof is complete. $\qquad\square$

## F. Proof of Proposition 2

*Proof.* Since $\mathbf{W}^{(t)}_{\text{whitened}} \neq \mathbf{W}^*$, the square of the trace of the gradient $\mathrm{Tr}[\mathbf{H}\mathbf{W}^{(t)}_{\text{whitened}}]^2$ must exceed some positive constant $C_G$, that is, $\mathrm{Tr}[\mathbf{H}\mathbf{W}^{(t)}_{\text{whitened}}]^2 > C_G$.

On the other hand, because:

1. The quadratic loss term $\mathrm{Tr}[(\mathbf{W}^{(t)}_{\text{whitened}})^\top\mathbf{H}\mathbf{W}^{(t)}_{\text{whitened}}]$ is upper-bounded on $\mathbb{R}^{n\times n}$, and

2. $\mathrm{Tr}[(\mathbf{W}_{\text{whitened}}^{(t)})^\top \mathbf{H}\mathbf{W}_{\text{whitened}}^{(t)}] \neq 0$ (due to $\mathbf{W}_{\text{whitened}}^{(t)} \neq \mathbf{W}^*$),

we have that there exists a positive number $0 < C_{\mathcal{L}}$ such that

$$0 < \mathrm{Tr}[(\mathbf{W}_{\text{whitened}}^{(t)})^\top \mathbf{H}\mathbf{W}_{\text{whitened}}^{(t)}] < C_{\mathcal{L}}.$$

Finally, since the norm of $\mathbf{H}$ is upper bounded, its trace must also be upper bounded by some constant $C_H$. Therefore, putting everything together, we have:

$$Q > \frac{C_G^2}{C_{\mathcal{L}} C_H}.$$

This inequality holds even as the condition number $\kappa \to +\infty$. $\qquad\square$

## G. Proof of Proposition 3

To prove Proposition 3, we first generalize existing work on the convergence rate lower bound (via contraction factor) of gradient descent and Adam (we only consider $\beta_2 = 1$) under the same setting:

**Theorem 3** (Contraction factor lower bound for gradient descent, generalized based on Zhang et al. (2024a)). *Consider the optimization problem in Equation (9). Let $\mathbf{W}_{GD}^t$ be the output of GD after $t$ steps. Then, for any step size $\eta$, there exists an initial condition such that the following lower bound on the contraction rate holds:*

$$\mathcal{L}(\mathbf{W}_{GD}^{t+1}) - \mathcal{L}^* \geq \left(1 - \frac{2}{\kappa + 1}\right)\left(\mathcal{L}(\mathbf{W}_{GD}^t) - \mathcal{L}^*\right),$$

*where $\mathcal{L}^* = \mathcal{L}(\mathbf{W}^*)$. Furthermore, under optimal $\eta = \frac{2}{\lambda_1 + \lambda_m}$, the bound becomes tight regardless of the settings of $\mathbf{H}$, where $\lambda_1$ and $\lambda_m$ are the largest and smallest eigen values of $\mathbf{H}$, respectively.*

*Proof.* The proposition 1 in Zhang et al. (2024a) has shown that the lower bound holds for diagonal positive definite Hessian $\mathbf{H}$. To show that the lower bound holds for a general positive definite Hessian $\mathbf{H}$ we will reformulate the problem to align with the setup in diagonal case (Prposition 1 of Zhang et al. (2024a)).

First, for any positive definite Hessian $\mathbf{H}$, we can perform an eigen decomposition $\mathbf{H} = \mathbf{U}\mathbf{S}\mathbf{U}^\top$, where $\mathbf{U}$ is an orthogonal matrix and $\mathbf{S}$ is a diagonal matrix containing the eigenvalues of $\mathbf{H}$. Define a change of variables $\mathbf{Z} = \mathbf{U}^\top \mathbf{W}$. Then, the optimization problem becomes

$$\mathcal{L}(\mathbf{Z}) = \frac{1}{2}\mathrm{Tr}(\mathbf{Z}^\top \mathbf{S}\mathbf{Z}),$$

which reduces the problem to the case of a diagonal $\mathbf{H}$ with condition number $\kappa = \frac{\lambda_1}{\lambda_m}$, where $\lambda_1$ and $\lambda_m$ are the largest and smallest eigenvalues of $\mathbf{H}$, respectively.

Thus, by applying Proposition 1 of Zhang et al. (2024a) to this transformed problem, we conclude that there exists initial point such that the lower bound on the contraction rate

$$\mathcal{L}(\mathbf{W}_{GD}^{(t+1)}) - \mathcal{L}^* \geq \left(1 - \frac{2}{\kappa + 1}\right)\left(\mathcal{L}(\mathbf{W}_{GD}^{(t)}) - \mathcal{L}^*\right)$$

holds for the transformed variables $\mathbf{Z}$ and, equivalently, for the original variables $\mathbf{W}$ since the condition number is preserved under orthogonal transformations.

Therefore, the lower bound for gradient descent applies to any general positive definite Hessian $\mathbf{H}$ provided the condition number $\kappa$ remains unchanged.

Finally, under the optimal step size $\eta = \frac{2}{\lambda_1 + \lambda_m}$, the bound becomes tight regardless of the settings of $\mathbf{H}$. This is achieved by selecting $\eta$ to minimize the contraction factor, aligning with well-known results regarding the optimal convergence rate of gradient descent on quadratic objectives (Nesterov, 2013).

This completes the proof of Theorem 3. $\qquad\square$

**Corollary 1** (Lower bound on Adam ($\beta_2 = 1$)). *Consider the optimization problem in Equation* (9). *Assume the weight initialization* $\mathbf{W}^0$ *assigned zero probability to any set of zero Lebesgue measure in* $\mathbb{R}^{m \times n}$. *Let* $\mathbf{W}_{Adam}^{(t)}$ *be the parameter after $t$ iterations of Adam with hyperparameters* $\beta_1 = 0$ *and* $\beta_2 = 1$. *Then, for any step size $\eta$, the following lower bound on the contraction rate holds:*

$$\mathcal{L}(\mathbf{W}_{Adam}^{(t+1)}) - \mathcal{L}^* \geq \left(1 - \frac{2}{\kappa'(\mathbf{W}^0) + 1}\right) \left(\mathcal{L}(\mathbf{W}_{Adam}^{(t)}) - \mathcal{L}^*\right),$$

*where* $\kappa'(\mathbf{W}^0)$ *is the* $\mathbf{W}^0$*-dependent condition number of the preconditioned Hessian* $diag(|\mathbf{HW}^0|^{-1})\mathbf{H}$, *and* $\mathcal{L}^* = \mathcal{L}(\mathbf{W}^*)$.

*Proof.* The update rule of Adam with $\beta_1 = 0$ and $\beta_2 = 1$ is given by Zhang et al. (2024a):

$$\mathbf{W}_{Adam}^{(t+1)} = \mathbf{W}_{Adam}^{(t)} + \eta \, diag(|\mathbf{HW}^{(0)}|^{-1}) \, \mathbf{HW}_{Adam}^{(t)}.$$

This can be interpreted as gradient descent with a preconditioned Hessian matrix $diag(|\mathbf{HW}^{(0)}|^{-1})\mathbf{H}$. By applying Theorem 3, we conclude that the contraction rate for Adam under these settings satisfies the lower bound:

$$\mathcal{L}(\mathbf{W}_{Adam}^{(t+1)}) - \mathcal{L}^* \geq \left(1 - \frac{2}{\kappa + 1}\right) \left(\mathcal{L}(\mathbf{W}_{Adam}^{(t)}) - \mathcal{L}^*\right),$$

where $\kappa$ is the condition number of the Hessian matrix $\mathbf{H}$.

Therefore, the proof is complete. $\qquad\square$

Next, We extend our Theorem 2 to block-diagonal Hessian case to prepare for discussions on group learning rates when comparing to Adam.

**Corollary 2** (**Upper Bound Convergence Rate of SWAN**). *Consider the same quadratic loss function* $\mathcal{L}(\mathbf{W}) = \frac{1}{2}Tr(\mathbf{W}^\top \mathbf{HW})$ *with* $\mathbf{H}$ *being block-diagonal. That is,* $\mathbf{H} = diag(\mathbf{H}_1, \mathbf{H}_2, \ldots, \mathbf{H}_L)$, *where each* $H_l \in \mathbb{R}^{m_l \times n_l}$ *is a positive definite matrix for* $l = 1, 2, \ldots, L$, *and* $\sum_{l=1}^L m_l = m$ *and* $\sum_{l=1}^L n_l = n$. *Assume the initialization distribution of* $\mathbf{W}^{(0)}$ *assignes zero probability to any zero measure set in* $\mathbb{R}^{m \times n}$. *Let* $\mathbf{W}_{whitened}^{(t)}$ *be the parameter matrix after $t$ iterations of the SWANoptimizer defined in Theorem 2, with learning rate $\eta$. Then, under the conditions that* [8]:

$$\|\mathbf{H}_l \mathbf{W}_l^{(t)}\|_1^2 - Tr(\mathbf{H}_l) \cdot Tr\left((\mathbf{W}_l^{(t)})^\top \mathbf{H}_l \mathbf{W}_l^{(t)}\right) \cdot \frac{2\lambda_{l,m_l}}{\lambda_{l,1} + \lambda_{l,m_l}} > 0,$$

*where* $\lambda_{l,1}$ *and* $\lambda_{l,m_l}$ *are the largest and smallest singular value of* $\mathbf{H}_l$, *respectively; then there exists a proper learning rate $\eta$ such that: with probability 1, the loss satisfies:*

$$\mathcal{L}(\mathbf{W}_{whitened}^{(t+1)}) - \mathcal{L}^* < \max_{l \in [L]} \left(1 - \frac{2}{\kappa_l + 1}\right) \left(\mathcal{L}(\mathbf{W}_{whitened}^{(t)}) - \mathcal{L}^*\right), \tag{19}$$

*where* $\kappa_l$ *is the condition number of* $\mathbf{H}_l$.

*Proof.* Applying the arguments in the proof of Theorem 2 to each block $l$, we have (for simplicity, we will drop the subscript "whitened" when there is no confusion):

$$\frac{\mathcal{L}(\mathbf{W}_l^{(t+1)}) - \mathcal{L}^*}{\mathcal{L}(\mathbf{W}_l^{(t)}) - \mathcal{L}^*} = 1 - \frac{\mathcal{L}(\mathbf{W}_l^{(t)}) - \mathcal{L}(\mathbf{W}_l^{(t+1)})}{\mathcal{L}(\mathbf{W}_l^{(t)}) - \mathcal{L}^*}$$

$$= 1 - \frac{\eta \|\mathbf{H}_l \mathbf{W}_l^{(t)}\|_1 - \frac{\eta^2}{2} Tr(\mathbf{H}_l)}{\frac{1}{2} Tr\left[(\mathbf{W}_l^{(t)})^\top \mathbf{H}_l \mathbf{W}_l^{(t)}\right]}.$$

---

[8]Note that according to Proposition 2, this is always achievable when $\mathbf{H}_l$ is poorly-conditioned ($\frac{\lambda_{l,d_l}}{\lambda_{l,1}}$ is small enough). The lower interval above converges to zero, and one can simply pick e.g., $\eta = \min_{l \in [L]} \frac{Tr(\mathbf{H}_l \mathbf{W}_l^{(t)})}{Tr(\mathbf{H}_l)}$.

It is straightforward to verify via the quadratic formula that if one chooses $\eta$ satisfying:

$$\frac{\|\mathbf{H}_l\mathbf{W}_l^{(t)}\|_1 - \sqrt{\|\mathbf{H}_l\mathbf{W}_l^{(t)}\|_1^2 - \mathrm{Tr}(\mathbf{H}_l) \cdot \mathrm{Tr}((\mathbf{W}_l^{(t)})^\top \mathbf{H}_l\mathbf{W}_l^{(t)}) \cdot \frac{2\lambda_{l,m_l}}{\lambda_{l,1}+\lambda_{l,m_l}}}}{\mathrm{Tr}(\mathbf{H}_l)} < \eta$$

$$< \frac{\|\mathbf{H}_l\mathbf{W}_l^{(t)}\|_1 + \sqrt{\|\mathbf{H}_l\mathbf{W}_l^{(t)}\|_1^2 - \mathrm{Tr}(\mathbf{H}_l) \cdot \mathrm{Tr}((\mathbf{W}_l^{(t)})^\top \mathbf{H}_l\mathbf{W}_l^{(t)}) \cdot \frac{2\lambda_{l,m_l}}{\lambda_{l,1}+\lambda_{l,m_l}}}}{\mathrm{Tr}(\mathbf{H}_l)},$$

then we have:

$$\frac{\frac{1}{2}\mathrm{Tr}\left[(\mathbf{W}_l^{(t)})^\top \mathbf{H}_l\mathbf{W}_l^{(t)}\right]}{\eta\|\mathbf{H}_l\mathbf{W}_l^{(t)}\|_1 - \frac{\eta^2}{2}\mathrm{Tr}(\mathbf{H}_l)} < \frac{\kappa_l + 1}{2}.$$

Rearranging, we obtain:

$$\frac{\mathcal{L}(\mathbf{W}_l^{(t+1)}) - \mathcal{L}^*}{\mathcal{L}(\mathbf{W}_l^{(t)}) - \mathcal{L}^*} < 1 - \frac{2}{\kappa_l + 1}.$$

Since $\mathbf{H}$ is block-diagonal, the updates for each block $l$ are independent. Summing the loss over all blocks, we obtain:

$$\mathcal{L}(\mathbf{W}_{\mathrm{whitened}}^{(t+1)}) - \mathcal{L}^* = \sum_{l=1}^{L}\left(\mathcal{L}(\mathbf{W}_l^{(t+1)}) - \mathcal{L}^*\right) < \sum_{l=1}^{L}\left(1 - \frac{2}{\kappa_l + 1}\right)\left(\mathcal{L}(\mathbf{W}_l^{(t)}) - \mathcal{L}^*\right).$$

Taking the maximum contraction factor across all blocks:

$$\mathcal{L}(\mathbf{W}_{\mathrm{whitened}}^{(t+1)}) - \mathcal{L}^* < \max_{l\in[L]}\left(1 - \frac{2}{\kappa_l + 1}\right)\sum_{l=1}^{L}\left(\mathcal{L}(\mathbf{W}_l^{(t)}) - \mathcal{L}^*\right)$$

$$= \max_{l\in[L]}\left(1 - \frac{2}{\kappa_l + 1}\right)\left(\mathcal{L}(\mathbf{W}_{\mathrm{whitened}}^{(t)}) - \mathcal{L}^*\right).$$

Thus, the overall contraction factor for the SWANoptimizer is:

$$\rho_{\mathrm{SWAN}} = \max_{l\in[L]}\left(1 - \frac{2}{\kappa_l + 1}\right).$$

$\square$

Finally, we are ready to prove Proposition 3:

**Proposition 2** (`GradWhitening` **with single lr vs Adam with tuned group lr**). *Consider the optimization problem Equation* (9). *Assume* $\mathbf{H}$ *is block-diagonal, i.e.,* $\mathbf{H} = \mathrm{diag}(\mathbf{H}_1, \mathbf{H}_2, \ldots, \mathbf{H}_L)$, *where each* $\mathbf{H}_l \in \mathbb{R}^{m_l \times m_l}$ *is a positive definite matrix for* $l = 1, 2, \ldots, L$, *and* $\sum_{l=1}^{L} m_l = m$. *Assuming for* `GradWhitening` *we use one global learning rate for all parameters; and for Adam, we use the optimally chosen group learning rate* $\eta_l$ *and initial condition* $w_0$ *for each block* $\mathbf{H}_l$. *Assume either if i) certain regularity conditions are met (see proof in Appendix), or ii), if* $\mathbf{H}$ *is poorly-conditioned (its condition number is large enough). Then: regardless of its initialization,* `GradWhitening` *with a properly chosen learning rate will still have a strictly better convergence speed (i.e., smaller contraction factor) across all blocks* $l \in [L]$ *than Adam* ($\beta_1 = 0, \beta_2 = 1$) *under optimal group-wise learning rates and initial condition.*

*Proof.* For simplicity, we will drop the subscript "whitened" when there is no confusion. Let $\kappa_l'(\mathbf{W}_l^{(0)})$ denote the $\mathbf{W}_l^{(0)}$-dependent condition number of the $l$-th block preconditioned Hessian $\mathrm{diag}\left(\left|\mathbf{H}_l\mathbf{W}_l^{(0)}\right|^{-1}\right)\mathbf{H}_l$. Let $\lambda_{l,m_l}(\mathbf{W}_l^{(0)})$ and $\lambda_{l,1}(\mathbf{W}_l^{(0)})$ be the smallest and largest eigenvalues of $\mathrm{diag}\left(\left|\mathbf{H}_l\mathbf{W}_l^{(0)}\right|^{-1}\right)\mathbf{H}_l$, respectively. Then,

**Case 1:** Under the conditions that:

1. **Existence of roots:** $\forall l \in [L]$,

$$\|\mathbf{H}_l \mathbf{W}_l^{(t)}\|_1^2 - \text{Tr}(\mathbf{H}_l) \cdot \text{Tr}\left((\mathbf{W}_l^{(t)})^\top \mathbf{H}_l \mathbf{W}_l^{(t)}\right) \cdot \frac{2\lambda_{l,m_l}(\mathbf{W}_l^{(0)})}{\lambda_{l,1}(\mathbf{W}_l^{(0)}) + \lambda_{l,m_l}(\mathbf{W}_l^{(0)})} > 0,$$

and

2. **Overlap condition:**

$$\min_{l \in [L]} \frac{\text{Tr}(\mathbf{H}_l \mathbf{W}_l^{(t)}) + \sqrt{\text{Tr}(\mathbf{H}_l \mathbf{W}_l^{(t)})^2 - \text{Tr}(\mathbf{H}_l) \cdot \text{Tr}\left((\mathbf{W}_l^{(t)})^\top \mathbf{H}_l \mathbf{W}_l^{(t)}\right) \cdot \frac{2\lambda_{l,m_l}(\mathbf{W}_l^{(0)})}{\lambda_{l,1}(\mathbf{W}_l^{(0)}) + \lambda_{l,m_l}(\mathbf{W}_l^{(0)})}}}{\text{Tr}(\mathbf{H}_l)}$$

$$> \max_{l \in [L]} \frac{\text{Tr}(\mathbf{H}_l \mathbf{W}_l^{(t)}) - \sqrt{\text{Tr}(\mathbf{H}_l \mathbf{W}_l^{(t)})^2 - \text{Tr}(\mathbf{H}_l) \cdot \text{Tr}\left((\mathbf{W}_l^{(t)})^\top \mathbf{H}_l \mathbf{W}_l^{(t)}\right) \cdot \frac{2\lambda_{l,m_l}(\mathbf{W}_l^{(0)})}{\lambda_{l,1}(\mathbf{H}_l^{(0)}) + \lambda_{l,m_l}(\mathbf{W}_l^{(0)})}}}{\text{Tr}(\mathbf{H}_l)}.$$

Then, there exists a global learning rate $\eta$, such that for all $l \in [L]$,

$$\frac{\mathcal{L}(\mathbf{W}_{\text{whitened}}^{(t+1)})_l - \mathcal{L}_l^*}{\mathcal{L}(\mathbf{W}_{\text{whitened}}^{(t)})_l - \mathcal{L}_l^*} < 1 - \frac{2}{\kappa_l'(\mathbf{W}_l^{(0)}) + 1} \leq \frac{\mathcal{L}(\mathbf{W}_{\text{Adam}}^{(t+1)})_l - \mathcal{L}_l^*}{\mathcal{L}(\mathbf{W}_{\text{Adam}}^{(t)})_l - \mathcal{L}_l^*}.$$

**Case 2:** If $\mathbf{H}$ is poorly-conditioned, i.e., $\frac{\lambda_{l,m_l}(\mathbf{W}_l^{(0)})}{\lambda_{l,1}(\mathbf{W}_l^{(0)})} \to 0$, then Proposition 2 asserts that the following term

$$\max_{l \in [L]} \frac{\text{Tr}(\mathbf{H}_l \mathbf{W}_l^{(t)}) - \sqrt{\text{Tr}(\mathbf{H}_l \mathbf{W}_l^{(t)})^2 - \text{Tr}(\mathbf{H}_l) \cdot \text{Tr}\left((\mathbf{W}_l^{(t)})^\top \mathbf{H}_l \mathbf{W}_l^{(t)}\right) \cdot \frac{2\lambda_{l,m_l}(\mathbf{W}_l^{(0)})}{\lambda_{l,1}(\mathbf{W}_l^{(0)}) + \lambda_{l,m_l}(\mathbf{W}_l^{(0)})}}}{\text{Tr}(\mathbf{H}_l)} \to 0,$$

and one can simply choose, for example, $\eta = \min_{l \in [L]} \frac{\text{Tr}(\mathbf{H}_l \mathbf{W}_l^{(t)})}{\text{Tr}(\mathbf{H}_l)}$. Under this choice of $\eta$, we still have

$$\frac{\mathcal{L}(\mathbf{W}_{\text{whitened}}^{(t+1)})_l - \mathcal{L}_l^*}{\mathcal{L}(\mathbf{W}_{\text{whitened}}^{(t)})_l - \mathcal{L}_l^*} < 1 - \frac{2}{\kappa_l'(\mathbf{W}_l^{(0)}) + 1} \leq \frac{\mathcal{L}(\mathbf{W}_{\text{Adam}}^{(t+1)})_l - \mathcal{L}_l^*}{\mathcal{L}(\mathbf{W}_{\text{Adam}}^{(t)})_l - \mathcal{L}_l^*}$$

for all $l \in [L]$. $\qquad\square$

## H. Proof of Proposition 1

*Proof.* First, define $\mathbf{V} \in \mathbb{R}^{M_C \times n}$ as $\mathbf{V} := \mathbf{U}_C^\top \mathbf{W}$, and consider the Hessian with respect to $\mathbf{V}$ instead of $\mathbf{W}$. Notice that although the loss function $\mathcal{L}$ is unknown, its first-order derivatives are known. Specifically, they are given by:

$$\frac{\partial \mathcal{L}}{\partial v_{lk}} = \dot{v}_{lk} = \mathbb{E}_{q=m}\left[g_{h_k} x_l\right] e^{\frac{1}{2} \sum_s v_{ls}^2}.$$

Therefore, the second-order derivatives, i.e., the Hessian matrix $\mathbf{H}(\mathbf{V})$, are:

$$\mathbf{H}(\mathbf{V})_{lk,l'k'} = \frac{\partial^2 \mathcal{L}}{\partial v_{lk} \partial v_{l'k'}} = \frac{\partial \left[\mathbb{E}_{q=m}\left[g_{h_k} x_l\right] e^{\frac{1}{2} \sum_s v_{ls}^2}\right]}{\partial v_{l'k'}} = \dot{v}_{lk} v_{l'k'} \delta_{ll'},$$

where $\delta_{ll'}$ is the Kronecker delta, which is 1 if $l = l'$ and 0 otherwise.

Based on Lemma B.6 in Zhao et al. (2024a), as $t \to \infty$, there exists an index subset $O_l \subset \{1, \ldots, M_C\}$ such that:

$$v_{l^*k} \gg v_{lk}, \quad \dot{v}_{l^*k} \gg \dot{v}_{lk}, \quad \forall l^* \in O_l, \ l \notin O_l, \ \forall k.$$

Consequently,

$$\mathbf{H}(\mathbf{V})_{l^*k,l^*k'} \gg \mathbf{H}(\mathbf{V})_{lk,l'k'}, \quad \forall l^* = l'^* \in O_l, \ l, l' \notin O_l, \ \forall k, k'.$$

After normalization, as $t \to \infty$, we have

$$\frac{\mathbf{H}(\mathbf{V})_{lk,l'k'}}{\sum_{l,l'} \mathbf{H}(\mathbf{V})_{lk,l'k'}} \xrightarrow{t \to \infty} \begin{cases} 1, & \text{if } l = l' \in O_l, \\ 0, & \text{otherwise.} \end{cases}$$

Reverting back to the $\mathbf{W}$ space, we have

$$\mathbf{H}(\mathbf{W}) = (\mathbf{I}_K \otimes \mathbf{U}_C)\mathbf{H}(\mathbf{V})(\mathbf{I}_K \otimes \mathbf{U}_C)^\top,$$

where $\otimes$ denotes the Kronecker product and $\mathbf{I}_K$ is the identity matrix of appropriate dimensions.

Therefore, for all $1 \leq s, s' \leq d$ and $1 \leq k, k' \leq n$, we obtain

$$\frac{\mathbf{H}(\mathbf{W})_{sk,s'k'}}{\sum_{s,s'} \mathbf{H}(\mathbf{W})_{sk,s'k'}} = \frac{\mathbf{H}(\mathbf{W})_{sk',s'k'}}{\sum_{s,s'} \mathbf{H}(\mathbf{W})_{sk',s'k'}} \quad \text{as } t \to \infty.$$

This holds for all $1 \leq s, s' \leq d$ and $1 \leq k, k' \leq n$. $\qquad\square$

**Remark** As pointed out by one of the reviewers, at equilibrium, $O_l$ contains only a single index. We acknowledge that during early training before convergence—multiple dominating indices may appear (as observed in numerical experiment in Figure 5), and similar structures are still present across the normalized diagonal blocks.

## I. Acceleration of Newton-Schulz iteration via diagonal substitution

### I.1. Algorithnms

Computing `GradWhitening` exactly can be expensive, as it involves solving the matrix square-root inverse. One option is to directly apply the Newton-Schulz variant of decorrelated batch normalization (Song et al., 2022; Li et al., 2018; Huang et al., 2019), which allows a more GPU-friendly estimation. This is given by (Song et al., 2022; Li et al., 2018):

$$\begin{cases} \mathbf{Y}_{k+1} = \frac{1}{2}\mathbf{Y}_k(3\mathbf{I} - \mathbf{Z}_k\mathbf{Y}_k) \\ \mathbf{Z}_{k+1} = \frac{1}{2}(3\mathbf{I} - \mathbf{Z}_k\mathbf{Y}_k)\mathbf{Z}_k \end{cases}$$

where $\mathbf{Y}_0 = \mathbf{G}\mathbf{G}^\top$, $\mathbf{Z}_0 = \mathbf{I}$. At convergence, `GradWhitening`$(\mathbf{G}) = \mathbf{Z}\mathbf{G}$ (Algorithm 2). However, estimating $(\mathbf{G}\mathbf{G}^\top)^{-1/2}$ with NS requires $\mathcal{O}(m^3)$ (assuming $m < n$) complexity. Here, we propose a heuristic scheme that has $\mathcal{O}(m^2)$ complexity to estimate square-root inverse:

$$\begin{cases} \mathbf{Y}_{k+1} = \frac{1}{2}\mathbf{Y}_k\text{Diag}(3\mathbf{I} - \mathbf{Z}_k\text{Diag}(\mathbf{Y}_k)) \\ \mathbf{Z}_{k+1} = \frac{1}{2}(3\mathbf{I} - \text{Diag}(\mathbf{Z}_k)\mathbf{Y}_k)\text{Diag}(\mathbf{Z}_k) \end{cases}$$

where $\text{Diag}(\cdot)$ returns a diagonal matrix that has the same diagonal elements as the input matrix. Basically, whenever we encounter matrix multiplication in NS iterations, we replace one of them by its diagonal approximation. We refer to this as the *NS with diagonal substitution* (NSDS) scheme.

Note that in the above standard presentation we have fixed that both NS and NSDS uses coefficients $= 0.5$ for $\mathbf{Y}$ updates and $\mathbf{Z}$ updates, respectively. In practice we may further tune these coefficients to compensate for short number of iterations (usually under 10).

### I.2. Experiment: LLM Gradient Condition Number Reduction

**Setup** In this synthetic experiment, we assessed the effectiveness of two whitening methods, Newton-Schulz and the proposed Newton-Schulz with diagonal substitution (NSDS) scheme, on gradient matrices obtained from LLM training. The exact `GradWhitening` operator results in matrices with optimal condition number ($= 1$); therefore, we hereby investigate

the matrix condition numbers of the processed gradients obtained from different methods. Specifically, both methods use 5 NS iterations with NS step size optimized. We train a 130M LLama model following the architecture setting of Section 5.1 on randomly generated sequences for 1000 steps, and take the MLP weights of a middle layer (we take the fifth layer without loss of generality) and use different methods to whiten the gradient matrices. We consider three methods: standard NS; NSDS; and SWAN with NSDS (that is, composing `GradNorm` with `NSDS-GradWhitening`). At each training step, the condition number reduction ratio of different method was calculated for both whitening methods (the higher the better). Note that for all methods, the gradients have been pre-normalized by its norm.

**Results**    Results are shown in Figure 6 **(a)**. We notice that NSDS alone (orange curve) is not sufficient to reach a good condition number reduction ratio. However, when combined with `GradNorm` (i.e., SWAN with NSDS, the green curve), its performance started to catch up and even outperform the standard NS method after 500 training steps. This show the effectiveness of the proposed scheme. One potential caveat that we spot is that the condition number produced by SWAN with NSDS is more noisy than the standard NS iteration; which might lead to improvements that will be addressed in future work.

**The significance of `GradNorm` and NSDS**    The results above highlight the importance of `GradNorm`. A potential question is whether `GradNorm`, when followed by `GradWhitening`, is merely a no-op that rescales the initial location of the NS iteration for better convergence. In fact, for all methods considered in Figure 6, the gradients have been pre-normalized by their (global) norm before being fed into each method. This re-scaling applied to all methods provides a negative answer to the question. To clarify further, we estimate the cosine similarities between the processed gradients produced by different pairs of methods. The results, shown in Figure 6 **(b)**, reveal the following: As the training iterations increase, the cosine similarity score between SWAN-NSDS and NS (denoted as $\cos(\text{SWAN-NSDS}, \text{NS})$) monotonically decreases to small values, indicating a near-orthogonal relationship. Both $\cos(\text{SWAN}, \text{NS})$ and $\cos(\text{SWAN}, \text{SWAN-NSDS})$ decrease over time, suggesting that both `GradNorm` and NSDS contribute to the orthogonality of $\cos(\text{SWAN-NSDS}, \text{NS})$. This demonstrates that the changes introduced by both `GradWhitening` and `NSDS-GradWhitening` are significant, rotating the update by a relatively large magnitude. This might also explain the observation in Section 5.1 that SWAN with NSDS behaves differently from other variants, showing slower early convergence but stronger long-term convergence.

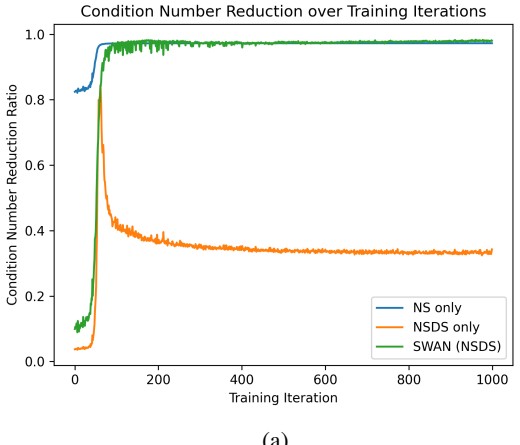
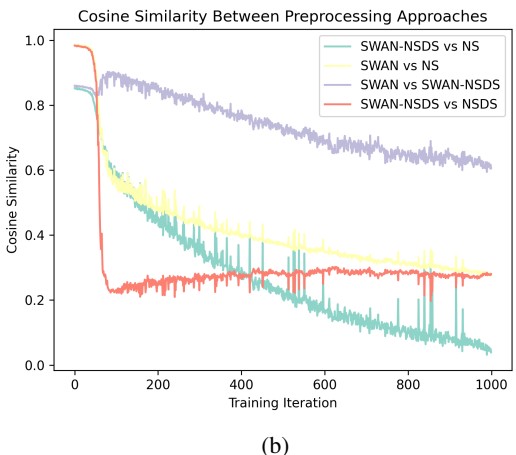

(a)                                                                                              (b)

Figure 6: Comparison between different whitening schemes. (a) Performance comparison by condition number reduction ratio (higher the better). (b) Cosine similarities between the whitened matrix produced by different preprocessing schemes, respectively.

## I.3. Ablation: effect of NSDS iterations

We examine the impact of the number of iterations of our Newton Schulz with Diagonal Substitution (NSDS) scheme. With SWAN‡, we compare the test PPL performance on 130M model. Results are shown in Table 4. As we can see, the improvement brought by additional NSDS iteration is marginal; hence in this paper, we only apply 2 iterations of NSDS.

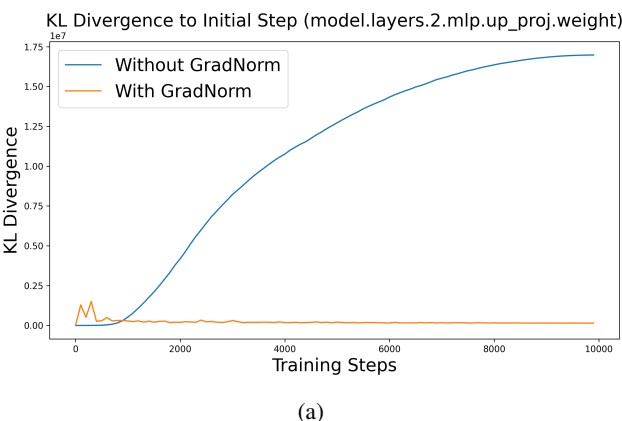
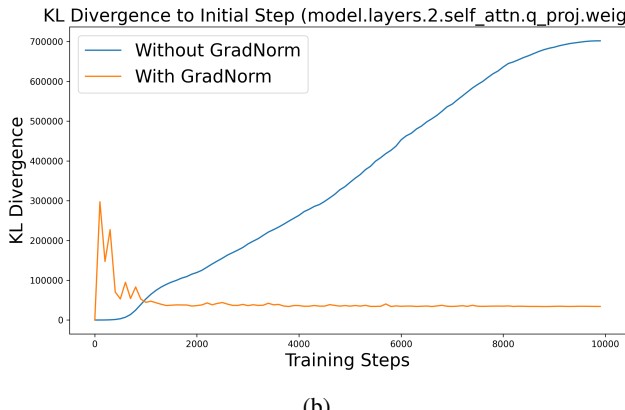

(a)            (b)

Figure 7: KL divergence comparison of gradient distributions against initial gradient distribution across training Steps. We use the projection weights in attention and MLP modules of the second layer as an example. The plots compare standard training with `GradNorm`-augmented training. Lower KL divergence values indicate greater stability in gradient distributions.

Table 4: Effect of NSDS iterations on test PPL. Results are obtain using FP32 precision.

| # NSDS iterations | Test PPL |
|---|---|
| 1 | - (LLM loss diverge) |
| 2 | 22.63 |
| 5 | 22.62 |
| 10 | 22.61 |

## I.4. Ablation: Precision

We compare the performance of different precisions of `GradWhitening` with NSDS scheme. As shown in Table 5, on 130M model (2 step NSDS iterations) ablation we show that BF16 can be used without major performance degrade compared to FP32.

Table 5: BF16 vs FP 32 NSDS

| NSDS Precision | Test PPL on 130M |
|---|---|
| BF16 | 22.61 |
| FP32 | 22.63 |

## J. Additional Experiments

### J.1. Empirical verification of theoretical insights from Section 4 regarding `GradNorm` and `GradWhitening`

#### J.1.1. DOES `GradNorm` STABILIZE GRADIENT DISTRIBUTIONS OF SGD?

To examine whether `GradNorm` stabilizes the distribution of the stochastic gradients as suggested by Theorem 1, we conduct controlled experiments using a scaled-down LLaMA-based model (about 10 million parameters) (Lialin et al., 2023), trained on the C4 dataset. Our goal is to measure how `GradNorm` affects the distribution of stochastic gradients over multiple training steps. Specifically, we employ a small-scale LLaMA-based model with approximately 10 million parameters (Lialin et al., 2023). Training is conducted on the C4 dataset (Raffel et al., 2020).

**Baselines** We compare:

- **Standard training**: This uses an SGD optimizer with a learning rate of $5 \times 10^{-4}$ and a linear learning rate scheduler, including a 10% warm-up of total training steps (10,000 steps).

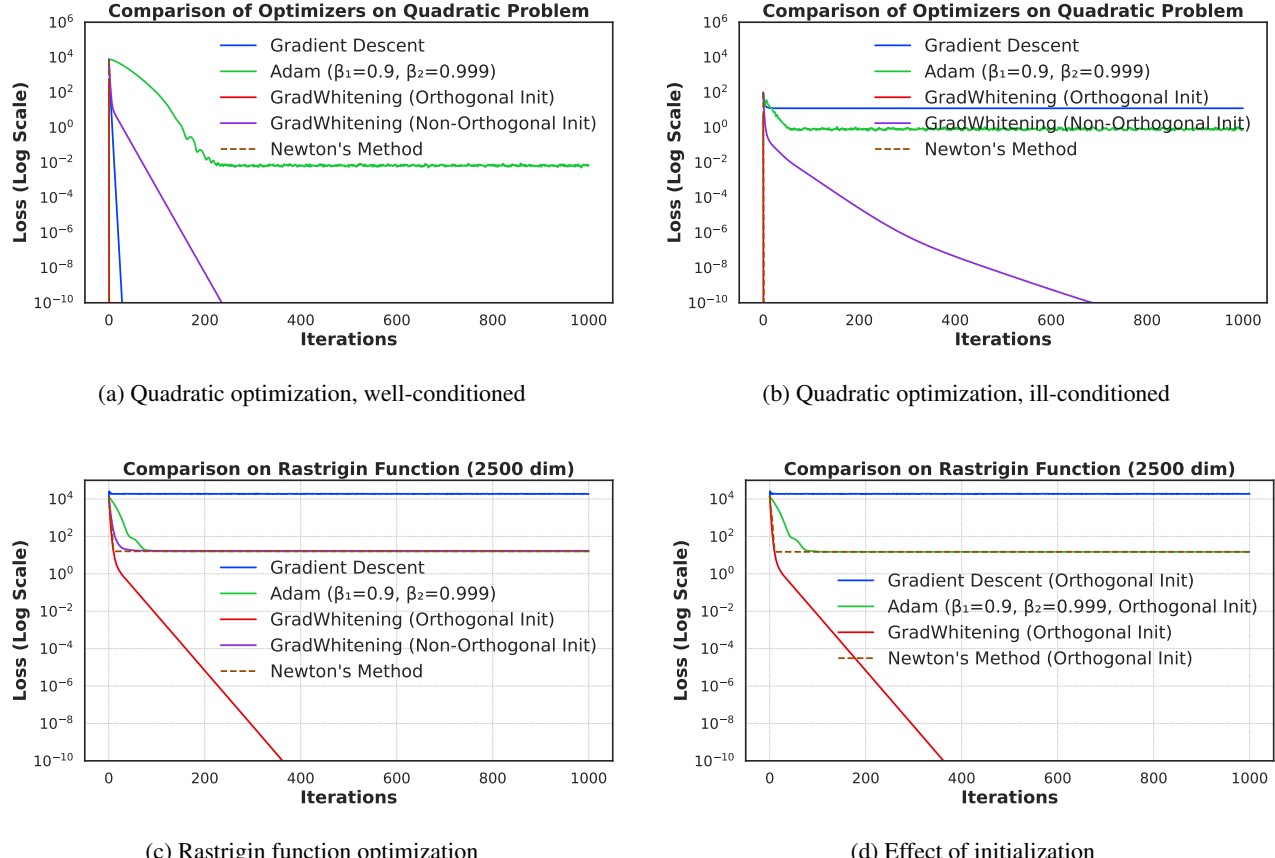

(a) Quadratic optimization, well-conditioned

(b) Quadratic optimization, ill-conditioned

(c) Rastrigin function optimization

(d) Effect of initialization

Figure 8: Comparison of convergence rate of different methods on quadratic and non-convex optimization problems. (a): 2500-dimensional quadratic optimization with well-conditioned $\mathbf{H}$. (b): 2500-dimensional quadratic optimization with ill-conditioned $\mathbf{H}$ (c): 2500-dimensional Rastrigin function optimization. (d): 2500-dimensional Rastrigin function optimization, but forcing all methods to use the same orthogonal initial location.

- `GradNorm`-**processed training**: This applies `GradNorm` to pre-process the stochastic gradient before the parameter update. All other settings match the standard training baseline.

**Methodology**    We measure gradient statistics in the presence of mini-batch noise. At step $t = 0$, we sample 16 additional mini-batches (batch size 64 each) and compute the mean and standard deviation of the corresponding raw or `GradNorm` gradients in each batch, and obtain the approximated initial gradient distribution. After each training step of baseline methods, we perform the same procedure and calculate the Kullback-Leibler divergence between the resulting gradient distributions and the initial gradient distributions. This process tracks how the gradient distribution changes over time.

**Results**    Figure 7 shows the KL divergence of gradient distributions for standard and `GradNorm`-augmented training, relative to the corresponding initial approximated distributions. Apart from early spikes, `GradNorm` reduces fluctuations in the gradient distribution throughout training.

### J.1.2. DOES `GradWhitening` COUNTERACTS LOCAL CURVATURE AND PROVIDE FAST CONVERGENCE ON ILL-CONDITIONED PROBLEMS?

This subsection evaluates the optimization performance of gradient descent when combined with `GradWhitening`. We use three classic problem settings:

- **High-dimensional quadratic optimization.** A quadratic problem of the form in Equation (9), where $\mathbf{W} \in \mathbb{R}^{50 \times 50}$.

- **Ill-conditioned quadratic optimization.** Same setup as above, but with a deliberately chosen ill-conditioned $\mathbf{H}$.

- **Non-convex optimization with multiple local optima.** We use the multivariate Rastrigin function:

$$f(\mathbf{W}) = m^2 \mathbf{A} + \frac{1}{2}\text{Tr}[\mathbf{W}^\top \mathbf{W}] - \mathbf{A}\sum_{ij}\cos(2\pi W_{ij}),$$

where $\mathbf{W}$ is an $m \times m$ matrix and $m = 50$. This function has $10^{m^2}$ possible local optima.

**Baselines**  We compare five methods on all three problems: gradient descent (GD) with the theoretical optimal learning rate in Theorem 3, Adam with $\beta_1 = 0.9$, $\beta_2 = 0.999$ and a hand-tuned learning rate, Newton's method with a tuned learning rate, and two `GradWhitening`-based variants (with and without orthogonal initialization). This is to verify, under orthogonal initialization, `GradWhitening`-processed GD behaves similarly to Newton's method, as discussed in Section 4.3 and Theorem 2. All methods share the same initialization, except for the orthogonal `GradWhitening` variant, which projects the initial parameters onto an orthogonal matrix.

**Results**  From Figure 8 (a)–(c), we summarizes the following outcomes:

- **Quadratic problems (Figure 8 (a) and (b)).** `GradWhitening` with orthogonal initialization and Newton's method converge to optimum in one step, aligning with the theoretical predictions in Section 4.3 and Theorem 2.

- **Well- vs. ill-conditioned cases (Figure 8 (a) and (b)).** In the well-conditioned setting (a), standard GD outperforms both Adam and `GradWhitening` (non-orthogonal initialization). In the ill-conditioned setting (b), `GradWhitening` (non-orthogonal initialization) outperforms GD by a large margin, while GD experiences slow convergence.

- **Comparison with Adam (Figure 8 (a)–(c)).** In all three settings, `GradWhitening` with non-orthogonal initialization consistently outperforms Adam, consistent with Proposition 3.

- **Rastrigin function (Figure 8 (c)).** On this non-convex problem, `GradWhitening` performs comparably to Newton's method, with or without orthogonal initialization.

- **Effect of initialization (Figure 8 (d))** Furthermore, we force all methods to share the same orthogonalized initialization. As shown by the result, `GradWhitening` GD with orthogonal initialization still consistently outperforms all baselines, confirming that this initialization is only beneficial to `GradWhitening` GD among all baselines.

## J.2. Is the speed-up Multiplicative or Additive?

A key question in assessing speedup factors is whether the improvement over Adam is *multiplicative* or *additive*. A multiplicative speedup implies that the optimizer's relative advantage remains proportionally consistent over time, while an additive speedup suggests a less desired constant step advantage. To investigate this, We focus on the SWAN-0 variant as an example, and explore these scenarios using two plots, the *speed-up ratio comparison* and the *perplexity comparison* (Figure 9), across different model sizes.

**Speedup Ratio Definition**  We define the *speedup ratio* $R(P)$ for a given perplexity (PPL) threshold $P$ as the ratio of the number of training steps Adam requires to reach a specific evaluation perplexity (PPL) to the number of required steps for SWAN:

$$R(P) = \frac{S_{\text{Adam}}(P)}{S_{\text{SWAN}}(P)}, \tag{20}$$

where $S_{\text{Adam}}(P)$ and $S_{\text{SWAN}}(P)$ are the training steps required by Adam and SWAN, respectively, to reach perplexity $P$. To test whether the speedup is additive, we compute a *counterfactual additive curve* by assuming SWAN gains a fixed step advantage $\Delta$ over Adam in early training (approximately the first 10%–20% of total steps):

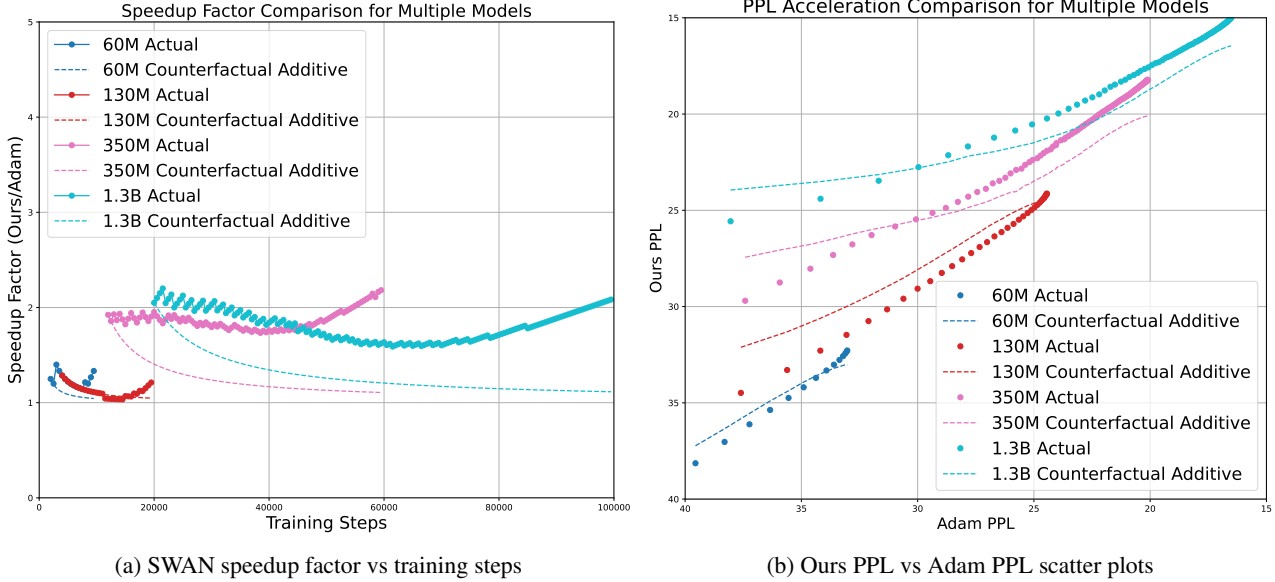

(a) SWAN speedup factor vs training steps      (b) Ours PPL vs Adam PPL scatter plots

Figure 9: Comparative analysis of SWAN and Adam optimizers: speedup ratios and perplexity metrics across various model sizes. **(a)** shows how SWAN reduces the number of training steps needed to achieve the same evaluation perplexity as Adam for models ranging from 60M to 1.3B parameters. A speedup ratio greater than one indicates that SWAN reaches target PPL values faster than Adam. **(b)** presents a direct comparison of perplexity scores between SWAN and Adam. In both plots, we also provide counterfactual additive curves (dashed lines), which displays hypothetical baselines that has a constant step advantages over Adam. Together, these plots highlight the nature of SWAN's speedup over Adam across different model scales.

**Counterfactual Additive Curve Estimation** To test whether the speedup is additive, we compute a *counterfactual additive curve* by assuming SWAN gains a fixed step advantage $\Delta$ over Adam in early training (approximately the first 10%–20% of total steps):

$$\Delta = \frac{1}{N} \sum_{i=1}^{N} \left( S_{\text{Adam}}(P_i) - S_{\text{SWAN}}(P_i) \right), \tag{21}$$

where $N$ is the number of PPL thresholds considered. We then use $\Delta$ to define the counterfactual additive speedup ratio:

$$R_{\text{additive}}(P) = \frac{S_{\text{Adam}}(P)}{S_{\text{Adam}}(P) - \Delta}, \tag{22}$$

and the counterfactual additive perplexity estimate:

$$\text{PPL}_{\text{additive}}(S) = \text{PPL}_{\text{Adam}} \left( S + \Delta \right). \tag{23}$$

This represents the expected perplexity of SWAN if it only consistently outperforms Adam by $\Delta$ steps.

**Results** We focus on the SWAN-0 variant as an example. Figure 9 compares SWAN-0's actual performance against the counterfactual additive curves. If its actual curves exceed these additive estimates, it indicates a tendency towards a multiplicative speedup, instead of the additive advantage. We summarize the observations for model sizes of 60M, 130M, 350M, and 1.3B parameters:

- For smaller models (60M and 130M), the actual speedup trajectories align closely with the additive baseline, indicating a primarily additive speedup.

- For larger models (350M and 1.3B), the actual curves rise noticeably above the additive estimates, suggesting a multiplicative speedup. This indicates that SWAN yields increasing efficiency gains as model size grows.

## J.3. On Adam hyperparameters sweep

In our main results presented in Section 5, the hyperparameters of Adam ($\beta_1$, $\beta_2$, $\epsilon$ etc) follows the setups of (Zhao et al., 2024a). Moreover, weight decay is not used. Below, we further fine-tune those hyperparameters and comapare with our method. We conducted an additional parameter sweep for lr, $\beta_1$, $\beta_2$, weight decay, and $\epsilon$ specifically for the 1B model scale, and obtained the following optimal values:

- Learning rate: 0.0007

- Betas: (0.9, 0.95)

- $\epsilon$: 1e-8

- Weight decay: 0.1 (we observed that a weight decay of 0.1 outperforms no weight decay, though we did not perform an exhaustive search)

We further trained the 1B model on 20B tokens using Adam with these optimal settings and our method. Apart from hyperparameters, the general training setup still follows (Zhao et al., 2024a). The following table summarizes the test loss (lower is better) at various training steps:

Table 6: Comparison of the test loss obtained during training when training 1B LLaMA with SWAN$^\ddagger$ v.s. 1B LLaMA under optimally tuned Adam.

| Method | 40K | 80K | 120K | 150K |
|---|---|---|---|---|
| Adam | 2.880 | 2.728 | 2.659 | 2.651 |
| SWAN$^\ddagger$ | **2.810** | **2.661** | **2.574** | **2.556** |

For our method, we still use the default parameter setting described in Appendix K. Our method still outperforms Adam by a large margin despite we spent much more compute to sweep the Adam hyperparameters.

# K. Implementation details

**General setup**    We describe the implementation setups for SWAN used in LLM pre-training tasks. To enable a more straightforward and comparable analysis, we simply replicate the setting of (Zhao et al., 2024a), under exactly the same model configs and optimizer hyperparameter configs, whenever possible. This includes the same model architecture, tokenizer, batch size, context length, learning rate scheduler, learning rates, subspace scaling, etc.

**Precision**    All baselines uses BF16 for model weights, gradients, and optimizer states storage. For all SWAN variants, we use BF16 for model weights and gradients. For the `GradWhitening` step of SWAN-0 and SWAN† we use FP32 to whiten the BF16 gradients and then convert it back to BF16. We found that this helps to improve traininig stability and performance. However, for the NSDS scheme of SWAN‡ we oberve that FP32 does not offer performance boost over BF16 (Appendix I.4), therefore we stick with BF16.

**Learning rate scheduling**    we use exactly the same scheduler as in (Zhao et al., 2024a), with the exception of SWAN-0, which does not require any learning rate warmup. Therefore, for SWAN-0, we directly start with maximum learning rate, and enter the learning rate decay phase, using the same decay parameters as (Zhao et al., 2024a).

**Reproducing baseline results**    Most baseline results are cited from respective papers (Zhao et al., 2024a; Zhu et al., 2024) as we share the exact same setup. We also tried to reproduce their results using the same opensourced code, and generally obtain slightly worse results for Galore, Apollo and Adam for larger models (350M and 1B, see Table 7). Therefore in the main paper we compare with both the official results and our reproduced results. For the reproduced results of Adam, we specify the details below as it was not disclosed in (Zhao et al., 2024a). We use same learning rate tuning procedure as suggested by (Zhao et al., 2024a) (i.e., performing grid search over $\{0.01, 0.005, 0.001, 0.0005, 0.0001\}$). We found that the optimal learning rates for Adam is 0.001. The only exception is that for a model of size 1.3B: as we already know that a

Table 7: Reproduced results.

|  | 130M | 350M | 1.3 B |
|---|---|---|---|
| Adam (reproduced) | 24.44 (0.75G) | 19.24 (2.05G) | 16.44 (7.48G) |
| Apollo-mini (reproduced) | 23.97 (0.43G) | 17.60 (0.93G) | 14.37 (2.98G) |
| Galore (reproduced) | 24.67 (0.57G) | 19.74 (1.29G) | 15.89 (4.43G) |
| $r$ of low-rank methods | 256 | 256 | 512 |
| Training Steps | 20K | 60K | 100K |

larger model requires smaller learning rates, we conduct a learning search for Adam over a smaller but more fine-grained grid of $\{0.001, 0.0007, 0.0005, 0.0003, 0.0001\}$. As a result, the optimal learning rate found for Adam on 1.3B is 0.0007. For other hyperparamters, we follow (Zhao et al., 2024a) where we use $\beta_1 = 0.9$, $\beta_2 = 0.99$, and no weight decay. Finally, one baseline that does not exist in the literature is the Momentum + `GradWhitening` (Muon-like optimizer without Nestrov acceleration) baseline; and we report our own results. We start from the default learning rates used by Muon (Jordan et al., 2024) and tuned them over a grid of $0.01, 0.02, 0.03, 0.04, 0.05$.

**SWAN Settings and hyperparameters**   Since SWAN utilizes matrix-level operations on gradients, it can only be applied to 2D parameters. Therefore, in our experiments, we only apply SWAN on all linear projection weights in transformer blocks. Similar to Galore (Zhao et al., 2024a), the rest of the non-linear parameters still uses Adam as the default choice. Therefore, we follow the learning rate setup of Galore, where we fix some global learning rate across all model sizes and all modules. Then, for the linear projection modules where SWAN is applied, we simply apply a scaling factor $\alpha$ on top of the global learning rate. For all SWAN variants, we adopt a *lazy-tuning approach* (hyperparameters are set without extensive search), as detailed below. This helps to reduce the possibility of unfair performance distortion due to excessive tuning.

- **SWAN-0** uses naive NS-iteration for whitening, disabled learning rate warmup, and use similar learning rates optimized for Adam. We fix the global learning rate to be the same as Adam, and fix $\alpha = 1$. The only exception is the 1.3 B case. This is because we observe that the optimal learning rate of Adam under 1.3B becomes smaller than 0.001, hence we also reduce the learning rate on SWAN, where we used $\alpha = 0.3$, resulting an effective learning rate of 0.0003. To summarize the hyperparameter of **SWAN-0** is set to be similar to Adam, without any tuning. This is to demonstrate the robustness of SWAN series optimizers and their capability to work out-of-the-box as a replacement for Adam.

- **SWAN†**, is the vanilla version of our method, in which we enabled learning rate warmup, and allowed the use of optimized learning rates that largely differ from Adam. We notice that SWAN allows larger learning rates than Adam. We use a global learning rate of 0.02, as well as the scaling factor $\alpha = 0.05$. This is selected by simply searching the learning rate over a constraint grid $\{0.01, 0.02, 0.05\}$, and then setting $\alpha = 0.05$ such that the effective learning rate is scaled back to 0.001. There is no guarantee that this heuristic rule is optimal; but we found that this usually does not make a run fail (e.g., with loss divergence).

- Finally, **SWAN‡**, the most efficient version of SWAN that employs the proposed NSDS scheme for fast whitening (section 3.2). Similar to **SWAN†**, we use the same global learning rate of 0.02, as well as the scaling factor $\alpha = 0.05$ across all model sizes. We suspect with more careful tuning, its performance can be significantly improved; however, this is out of the scope of the paper.

The configurations of `GradWhitening` is discussed next.

**Implementation of `GradWhitening`**   For `GradWhitening`, before N-S iteration, we further normalize its input matrix by its Frobenius norm. This is applied in all our ablation studies as well, regardless of whether the gradient is processed by `GradNorm`. For our proposed Newton-Schulz with Diagonal Substitution (NSDS) used in SWAN‡, we only run it for 2 steps across all model sizes. We found that using a step size $\beta \neq 0.5$ in the `GradWhitening` operator (Algorithm 2) can improve its convergence. We set $\beta = 0.4$. We found that NSDS is generally robust to $\beta$ as long as it is not too large; and our specific choice of parameters usually already gives satisfactory performance in LLM pretraining. For the naive N-S iteration used in SWAN-0 and SWAN†, we run 10 steps which is usually sufficient. We set $\beta = 0.8$. The naive NS is run in FP32 precision, on top of BF16 gradients and weights; while the NSDS is run in BF16.

**Computational Overhead.** Below, we discuss the computational overhead of Naive Newton-Schulz. In practice, we only run the naive Newton-Schulz for $\leq 10$ iterations, which corresponds to $\leq 50$ matrix multiplications. These matrix multiplications are in general GPU friendly, hence for the task of training LLMs, the batch size is the more dominant factor for compute. For example QWen 14B (Bai et al., 2023) has 4M batch size vs a a model dimension $d_{\text{model}} = 5120$, DeepSeek (Bi et al., 2024) 67B has 6M batch size vs $d_{\text{model}} = 8192$, and LLama 3 (Dubey et al., 2024) 405B has 4-16M batch size vs $d_{\text{model}} = 16384$. To estimate the computational overhead, assuming the N-S iteration involves approximately $50 \times d_{\text{model}}^3$ FLOPs. In contrast, the primary training cost scales with the batch size and is proportional to batch_size $\times d_{\text{model}}^2$ FLOPs. In those examples, the estimated computational overhead of Newton–Schulz is typically below $\leq 7\%$. A similar estimation has been given in (Jackson, 2023) ($\leq 5\%$), as well as in (Jordan et al., 2024) ($\leq 1\%$).

However, note that whether the above analysis hold for large scale, distributed LLM training is still unclear. The N-S iteration is after all a $\mathcal{O}(m^3)$ complexity operation and might need to scale as model size increases. As argued by (Essential AI, 2025), for MoE models the ratio of NS flops vs model flops can be much worse than dense models. This is exactly the motivations that drives us to explore alternative schemes.

