# OpenReview forum: "SWAN: SGD with Normalization and Whitening Enables Stateless LLM Training"
_ICML.cc/2025/Conference — ICML 2025 poster_

### Official Review · Reviewer_KRz9 · 2025-03-08

**Overall Recommendation:** 3

**Summary:**

This paper combines two classic technique normalization and whitening to improve SGD and achieves better performance than Adam and other optimizers while saving memory cost by not saving optimizer state. Theoretical insights are also provided to explain the effect of each technique.

**Claims And Evidence:**

The superior performance of SWAN is supported by the results in table 1. The authors also conduct ablation study to show that both modifications are necessary. I am not convinced by the theoretical analysis in section 4. See the theoretical part and weakness part for details.

**Essential References Not Discussed:**

The paper has included recent progress on designing new optimizers.

**Experimental Designs Or Analyses:**

The experiment design is clear, with three different version of SWAN to make a fair comparison with previous methods. I have a question for figure 6. When you say you compute mean of GradNorm gradient, do you compute the mean of GradNorm(G) or the original G but obtained from checkpoints trained with GradNorm update rule? If it is the mean of GradNorm(G), then the results sound less interesting because the distribution indeed should change less when you have a hard constraint on the scale of random variables.

**Methods And Evaluation Criteria:**

The proposed method combines two classic technique that are widely used by previous work. It is interesting that both techniques together provide a much better performance than each of them alone. The selected benchmark and models of different sizes are reasonable.

**Other Comments Or Suggestions:**

1. There is a typo in the definition of GradNorm (right part of line 146). You seem to miss a square term when defining $s$.
2. Another typo in theorem 1 when you define the standardized stochastic gradient.
3. It will be good to list all the assumptions rather than just saying inheriting from other paper.

**Other Strengths And Weaknesses:**

Strength
1. The paper proposed an efficient algorithm for GradWhitening. It is interesting that it converges very fast.

Weakness
1. Even though the authors try to give justification for the methods from the theoretical prospective, they only prove some results for each technique separately. It is unclear why combining them together can improve the results so much. Also each theoretical result will break when you add another operation. So I feel all the theoretical results are not so meaningful.

**Questions For Authors:**

1. Have you considered switch the order of gradnorm and gradwhitening?
2. Have you tried normalizing along another axis when doing gradnorm? I am curious whether we can get a similar analysis (theorem 1) for normalizing along the other axis. If not, why is one better than the other?
3. Do you have any insight on why combining the two techniques can improve performance over each of them own?
4. Question 1 in theoretical part.
5. Question 2 in theoretical part.

**Relation To Broader Scientific Literature:**

This paper follows previous work on designing a more efficient optimizer for training LLMs.

**Theoretical Claims:**

I checked the proof of theorem 2, proposition 2 and proposition 1. I feel the results are either trivial or their proofs lack detail for me to confirm correctness.

1. In the proof of proposition 2, can you explain the existence of C_G and C_L? If we just consider $H=diag(h_i)$ and $W_{whitened}^{(t)}=diag(w_i)$, then $Q=\frac{(\sum h_i w_i)^2}{(\sum h_i w_i^2)(\sum h_i)}$ can be arbitrarily small even with fixed $h_i$.
2. In the proof of proposition 1, can the subset $O_l$ have multiple elements? Is there any conclusion for the relationship between $H(V)\_{lk,lk’}$ and $H(V)_{l’k,l’k’}$ for both $l,l’ \in O_l$? Moreover, the convergence result of $H(V)$ after normalization can’t hold when the size of $O_l$ is larger than 1.

---

> ### Author Rebuttal · Authors · 2025-03-31
>
> **We thank the reviewer for their thorough and constructive feedback. We are grateful that the reviewer acknowledged the novelty, writing, empirical performance, and the overall significance of our work. Below, we address each point raised:**
>
>
> ---
>
> **1. On Combining GradNorm and GradWhitening:**
> - This is an excellent question.  Indeed, we found there is a theoretical explanation behind this, which in fact gave rise a class of more general algorithms than SWAN.  However, this insight is non-trivial and, due to space limitations, we plan to present a more comprehensive analysis in follow-up work. Below we briefly describe the high level idea:
>
> - To start with, given a gradient matrix $G$ with shape $m$ by $n$, GradNorm and GradWhitening can be interpreted as projection operators under specific norm constraints. The extended version of SWAN can be thus viewed as an iterative process:
> $$
> G \rightarrow \text{GradNorm}(G) \rightarrow \text{GradWhitening}(\text{GradNorm}(G)) \rightarrow \text{GradNorm}(\text{GradWhitening}(\text{GradNorm}(G))) \rightarrow \ldots
> $$
> until a fixed point is reached. In other words, it corresponds to performing steepest descent under multiple (non-Euclidean) norm constraints, as opposed to the standard SGD update which is steepest descent under a single Euclidean norm constraint.
> -  In theory, one could choose an arbitrary collection of norms—provided they satisfy certain theoretical properties—to guide the update. When \(G\) is nearly a square matrix, even a single iteration (as implemented in SWAN) is nearly sufficient.
> - In revision, we will present the main results as extended discussion.
>
> ---
>
> **2. Order of GradNorm and GradWhitening (Reviewer Question Q1):**
> From a fixed-point perspective above, the final result is invariant to the order as long as the iterative process converges to a multi-norm fixed point. Hence, the key is not the order but achieving convergence under the imposed norm constraints.
>
> ---
>
> **3. Normalizing Along Different Axes (Reviewer Question Q2):**
> We indeed experimented with normalizing along alternative axes. We found that normalizing along the row-wise direction (as in our current formulation) indeed yields superior performance. This is consistent with Theorem 1, which suggests that the primary source of gradient noise exhibits a row-wise scaling structure. We will include additional discussion and experimental evidence of this in the revised manuscript.
>
> ---
>
> **4. Regarding the Proof of Proposition 1:**
> The reviewer asks whether the subset $O_l$ can have multiple elements. In fact, at equilibrium, $O_l$ contains only a single index. We acknowledge that during early training—before convergence—multiple dominating indices may appear (as observed in numerical experiment in Figure 8), and similar structures are still present across the normalized diagonal blocks. We will clarify this point in the revision to make it explicit that our theoretical results assume convergence, at which point $O_l$ becomes a singleton.
>
> ---
>
> **5. Regarding the Proof of Proposition 2:**
> Our argument in Proposition 2 is not that the bound cannot be arbitrarily small, but rather that a small condition number does not necessarily lead to a diminished bound. This distinguishes our method from standard SGD, where such a decrease would be expected. We will revise our presentation to better articulate that our bound remains meaningful even when the gradient matrix is well-conditioned.
>
> ---
>
> **6. Typos and Presentation of Assumptions:**
> We thank the reviewer for pointing out typos and suggesting improvements in the presentation of our assumptions (e.g., in Theorem 1 and throughout the appendix). We will carefully revise the manuscript to correct these issues and to list all relevant assumptions explicitly.
>
> ---
>
> **Once again, we appreciate the reviewer's insightful comments, which help us improve both the theoretical exposition and empirical presentation of our work.**

---

> > ### Comment · Reviewer_KRz9 · 2025-04-02
> >
> > Thanks for the clarification of the theoretical proof. I will keep my score and look forward to the discussion of the fixed-point analysis.

---

### Official Review · Reviewer_urpW · 2025-03-13

**Overall Recommendation:** 3

**Summary:**

This paper introduces a new stateless optimizer SWAN (SGD with Whitening And Normalization) with the same performance as the Adam optimizer for LLM training. The author analyses that SGD with GradNorm and GradWhitening applied in tandem can minimize the condition number, stabilize gradient distributions across transformer block and coverage more robust to the local curvature condition. This paper evaluates the SWAN on LLM pre-trained tasks, where SWAN outperformed other optimizers, and analyses the effect and efficiency of SWAN.



## update after rebuttal:

I have read the response by the authors and my concerns are mostly addressed. I keep my score.

**Claims And Evidence:**

The claims are overall supported by theoretical or experimental evidence. The evidences are clear and convincing.
However, the motivation of applying normalization and whitening onto gradient is not very clear. Especially the best performance is usually obtained with around 2 iterations when using Newton’s iteration to obtain an approximated whitened/orthogonalized representation. What is the results when using more iterations? E.g. 5 iterations.

**Essential References Not Discussed:**

I think this paper should give credit to the normalized gradient method in training DNNs, e.g., the paper [1]

[1]Block-normalized Gradient Method: an Empirical Study for Training Deep Nerual Network. preprint arXiv:1707.04822

**Experimental Designs Or Analyses:**

The experiments are overall comprehensive. The results show that SWAN outperformed other optimizers. The authors conduct ablation experiments to figure out the effect of GradNorm and GradWhitening and why and how they help optimization. The author also analyses memory efficiency and throughput of SWAN.

**Methods And Evaluation Criteria:**

The methods and evaluation criteria are reasonable.

**Other Comments Or Suggestions:**

Typo problem: The punctuation after the equations are not unified.

**Other Strengths And Weaknesses:**

The paper is well organized.
However, the colored column in Figure 2 is confusing. It would be better if more detailed explanation is marked in this figure.

**Questions For Authors:**

1.	In Section2, the author hypothesize the additional history information in Adam is because the approach does not take into account the interactions and structures between different variables. Could the authors provide more evidence?
2.	According to my understanding, the gradient matrix G refers to the stochastic gradient of a particular weight matrix. The elements among columns of G are all variables. Therefore, how to understand whitening the gradient data in GradWhitening? May be the term “orthogonal the gradient vector” better?
3.	The experiments only involve C4 dataset. Could the advantage of SWAN preserve when it transfers to other tasks, for example vision or Multimodal tasks? (Even though this paper addresses for LLM training, it seems the method is general without considering what type data is)

**Relation To Broader Scientific Literature:**

The authors provide a new optimizer combining with normalization and whitening technology, which is effective and efficient in memory according to the experiments.

**Theoretical Claims:**

I do not find remarkable errors in theoretical claims. The claims are supported by analysis and previous studies. The theoretical analysis in this paper is somewhat solid, providing both practical considerations and dynamic analysis. One main concern is that the assumption 1 is too strong, and I donot think assumption 1 hold in practice.

---

> ### Author Rebuttal · Authors · 2025-03-30
>
> **We thank the reviewer for their insightful feedback and for acknowledging many of the strengths of our work. We address the specific points raised below:**
>
> **1, Iteration Count and Motivation for Whitening:**
> Regarding the number of iterations in the Newton–Schulz procedure, our ablation experiments in Appendix B show that the performance does not peak at 2 iterations; rather, increasing the iteration count (e.g., to 5) continues to improve the approximation accuracy of the whitening operator. We chose 2 iterations in our main experiments as a trade-off between computational cost and performance, while the improvement with additional iterations is clearly demonstrated in our supplementary results.
>
> For motivation for whitening, please refer to our response below.
>
> **2, Evidence for the claim that Adam needs historical information because it ignores non-diagonal interaction:**
> In Section 5.3 of our paper, we provide theoretical motivation for why Adam requires historical information—and why our method does not. Below, we rephrase the analysis from the natural gradient descent/Fisher information perspective. First, recall that the Fisher information matrix (FIM) is defined as $F = \mathbb{E}[gg^T]$ where $G$ is the gradient matrix and $g = \operatorname{vec}(G)$ is the flattened gradient (hence $F$ is $mn$ by $mn$.). Adam approximates the FIM using two key approximations:
> - It uses exponential moving averages (EMAs) over time to estimate the expectation $\mathbb{E}[\cdot]$.
> - It further adopts a diagonal approximation to $F$, thereby ignoring interactions between different parameters.
>
> In contrast, our gradient whitening operation is based on a block (identical) diagonal structural assumption:
> $$
> \tilde{F} = I \otimes M,
> $$
> where $\otimes$ is the Kronecker product,  $I$ is the $n$ by $n$ identity matrix and $M$ is a $m$ by $m$ matrix represents the FIM for each identical block diagonals. We can find the optimal $M$ by solving the optimization problem
> $$ \min_{\tilde{F} = I \otimes M}  ||\tilde{F} - F||^2_{Frobenius} $$
> the optimal solution for each block is given by the unbiased estimate
> $$ M \approxeq \frac{1}{n} \sum_{j=1}^n g_j g_j^T = GG^T $$
> with the summation taken over the columns of $G$ (each column corresponds to a "diagonal block" in the $F$). Then, we can use the optimal $\tilde{F}$ to perform natural gradient descent where the update is given by:
> $$\Delta  \operatorname{vec}(W) = \tilde{F}^{-1/2} g$$
> simplifying to matrix form, this is equivalent to
> $$\Delta  W \propto  M^{-1/2} G$$, which is exactly (up to scaling) $\text{GradWhitening}(G) = (GG^T)^{-1/2}G$.
>
> In plain language, we have shown that: because $\tilde{F}$ assumes each diagonal block is identical, one can estimate the full FIM using spatial information (i.e., average over blocks using $\frac{1}{n} \sum_{j=1}^n g_j g_j^T$ instead of over time) rather than relying on temporal averaging. This key insight allows the whitening operation to be derived as the solution to the FIM approximation problem under our block diagonal assumption. This analysis shows that our structural assumption permits bypassing the need for historical averaging.
>
> **3, Clarification on Terminology for the Whitening Operation:**
> We follow standard terminology from the literature (e.g., in Decorrelated Batch Normalization by Huang et al., 2018) where the operation$(GG^T)^{-1/2}G$
> is commonly referred to as “whitening”. Another reason is that performing natural gradient descent using the inverse square root of FIM is also referred to as whitening in the literature. (Adam can also be seen as a diagonal approximation of the inverse square root of FIM). We will add further clarification in the revised manuscript to ensure this point is unambiguous.
>
> **4, Generalization Beyond the LLMs:**
> Although our experiments focus on LLM pretraining with the C4 dataset, we expect that with minimal changes, similar benefits could be achieved in vision or multimodal tasks. We plan to explore these extensions in future work.
>
> **5, Additional Revisions:**
> We appreciate the reviewer’s detailed suggestions regarding the clarity of the appendix (including the presentation of assumptions from Theorem 1 of Tian et al., 2023), the labeling in Figure 2, and consistency in punctuation after equations. We will carefully revise these sections to improve readability and ensure consistency throughout the manuscript.
>
> **Once again, we thank the reviewer for their constructive comments, which will help us improve the quality and clarity of our paper in the final revision.**

---

### Official Review · Reviewer_A9LJ · 2025-03-16

**Overall Recommendation:** 3

**Summary:**

The paper proposes SWAN, an optimizer which is completely stateless. They claim that SWAN outperforms existing optimizers while also using lesser memory (since it is stateless). They support these claim by doing LLM pretraining experiments.

SWAN is similar to another previously proposed optimizer Muon the following changes: 1. They remove momentum, 2. They add column normalization and 3. In one of the versions of SWAN they propose a simplified version of Newton-Schultz iterations which are needed for matrix whitening.

**Claims And Evidence:**

I think the claim that SWAN outperforms Muon is problematic since their comparison is done at a small batch size (130K) while it is well know from prior works that the benefit of momentum emerges at larger batch sizes.

**Essential References Not Discussed:**

Most references have been discussed.

**Experimental Designs Or Analyses:**

NA

**Methods And Evaluation Criteria:**

Yes, except for the small batch size.

**Other Comments Or Suggestions:**

I think the paper would benefit from focusing on one main benefit and showing evidence for it so that no doubt remains. For example, the aforementioned use of small batch sizes. I would also recommend that the authors release the codebase so that it is easy to see details such as weight decay (which is not specified in the paper).

**Other Strengths And Weaknesses:**

The authors do not describe how they set hyperparameters such as weight decay.

**Questions For Authors:**

1. For Muon optimizers, was Adam used for first and last layer? (as in recommended in the referenced blogpost on muon).
2. Could the authors do experiments with 1 million batch size (with tune LRs). Any >100M sized model should suffice.
3. Could the authors share their codebase?

**Relation To Broader Scientific Literature:**

This paper adds to many new recent optimizers trying to improve on Adam in both speed and memory requirements.

**Theoretical Claims:**

Yes

---

> ### Author Rebuttal · Authors · 2025-03-29
>
> **We thank the reviewer for their detailed and constructive feedback. We address the main points raised below:**
>
> 1. **"The claim that SWAN outperforms Muon is problematic"**
>    - Our core contribution is to push the boundaries and demonstrate that: **it is possible to train LLMs matching the performance of Adam using a completely stateless optimizer**.  Our baseline “Momentum-GradWhitening” is included mainly for completeness, and whether SWAN can consistently outperform other existing non-Adam optimizers (such as shampoo/Muon/SOAP) is orthogonal to our contribution.
>
>     - While the reviewer notes that it is questionable whether SWAN outperforms Muon, it is important to emphasize that Muon is not a stateless optimizer and is primarily designed for wall-clock time acceleration. Our work thus highlights a viable extreme stateless alternative for memory-constrained settings, and this distinction is central to our contribution. We will clarify this distinction in the revision.
>
>     - Finally, we would like to note that compared to Muon, the most relevant baseline should in fact be the concurrent work of Apollo [2] (which is a low-rank/rank-1 optimizer) that also claims to match the performance of Adam. In our experiments, we have demonstrated that SWAN consistently outperforms Apollo.
>
> 2. **Experiments with Larger Batch Sizes:**
>    We have conducted additional experiments training a 130M model with a 1M batch size over 2B tokens. For the Adam baseline, we performed a learning rate sweep and found the optimal parameters to be **lr = 0.00075** and **betas = (0.9, 0.95)**. The results are summarized in the table below:
>
>    | Training Steps | Adam Val Loss | SWAN (SWAN$^\dagger$) Val Loss |
>    |----------------|---------------|-------------------------------|
>    | 500            | 4.1796        | 4.0755                        |
>    | 1.5K           | 3.483         | 3.477                         |
>    | 2.5K           | 3.398         | 3.370                         |
>
>    These results demonstrate that SWAN remains on par with or slightly better than Adam in terms of validation loss, even when training with a significantly larger batch size. This again confirms our core finding of "it is possible to train LLMs matching the performance of Adam using a completely stateless optimizer"
>
> 3. **"For Muon optimizers, was Adam used for first and last layer?"**
>    Yes, following standard practice (which predates both our work and Muon), Adam is used for the first and last layers in Muon. SWAN also employs Adam for these layers, aligning with practices used in other baselines such as Galore and Apollo.
>
> 4. **"Could the authors share their codebase?"**
>    We are fully committed to open-sourcing our codebase. We are actively working on a robust open-source release that will include detailed configurations as well as more new optimizers that were not included in our submissions
>
> 5. **Regarding hyperparameter settings of baselines:**
>    As stated in our paper, we strictly follow the experimental setup of Zhao et al. (2024a) [1]. Our experiment code is based on their implementation, and the configurations—including weight decay and other hyperparameters for all baselines—are directly taken using the config files that can be found in their open-source repository. We will further clarify these details in the revised version of the paper.
>
> **Once again, we thank the reviewer for their insightful comments and suggestions, which will help us further improve the manuscript.**
>
> [1] Zhao, Jiawei, et al. "Galore: Memory-efficient llm training by gradient low-rank projection." arXiv preprint arXiv:2403.03507 (2024).
>
> [2] Zhu, Hanqing, et al. "Apollo: Sgd-like memory, adamw-level performance." arXiv preprint arXiv:2412.05270 (2024).

---

### Official Review · Reviewer_G4T7 · 2025-03-16

**Overall Recommendation:** 3

**Summary:**

The paper proposed a "stateless" optimizer using gradient-normalization and gradient-whitening. The proposed method saves half memory over Adam and reaches 2x speedup. The idea is interesting and the writing is clear.

**Claims And Evidence:**

yes

**Essential References Not Discussed:**

see below

**Experimental Designs Or Analyses:**

yes

**Methods And Evaluation Criteria:**

yes

**Other Comments Or Suggestions:**

see blow

**Other Strengths And Weaknesses:**

see below

**Questions For Authors:**

Q1: In Figure 1,  12B tokens are not sufficient for 1B models. Does the advantage of SWAN maintain if we train more tokens?

Q2: In Figure 1 (c), does SGD refers to SGD with momentum or without momentum? Further, to make a clearer comparison with existing works, please also include other memory-efficient methods (e.g., Adam-mini, Muon) in the bar plot of Figure 1 (c).

Q3: In Algorithm 1 SWAN, what if keep track of a 1st-order momentum M and apply the  GradNorm() and GradWhitening() to M, instead of G? Does it bring extra acceleration?

Q4: [1]  studied why SGD performs poorly on Transformers.  Please discuss [1] as a motivation to re-design new stateless methods.

[1] Zhang, Y., Chen, C., Ding, T., Li, Z., Sun, R., & Luo, Z. (2024). Why transformers need adam: A hessian perspective. Advances in Neural Information Processing Systems, 37, 131786-131823.

**Relation To Broader Scientific Literature:**

no

**Theoretical Claims:**

yes

---

> ### Author Rebuttal · Authors · 2025-03-29
>
> **We thank the reviewer for their careful reading and for the positive comments regarding the clarity of our writing and the significance of our contribution. We address each question below:**
>
> **Q1:**
> Regarding the sufficiency of 12B tokens for 1B models, we have conducted additional experiments training a 1B model for 20B tokens. For the Adam baseline, we performed an extensive parameter sweep and found that the optimal parameters were **lr = 0.0007** and **betas = (0.9, 0.95)**. Using these settings, we compared the performance of Adam and SWAN (using the SWAN$^\ddagger$ setting with the same learning rate as Adam). The table below summarizes the training loss at various steps:
>
> | Training Steps | Adam Loss | SWAN Loss |
> |----------------|-----------|-----------|
> | 20K            | 3.054     | 2.989     |
> | 40K            | 2.880     | 2.810     |
> | 60K            | 2.792     | 2.726     |
> | 80K            | 2.728     | 2.661     |
> | 100K           | 2.681     | 2.609     |
> | 120K           | 2.659     | 2.574     |
> | 150K           | 2.651     | 2.556     |
>
> At the end of training, SWAN achieved a roughly **1.8× speedup** compared to Adam while reaching lower validation loss values. These detailed comparisons confirm that the advantage of SWAN persists even when training with more tokens and under a longer training schedule. (In fact, in this run, the extra steps Adam must take to reach SWAN's performance keep growing.)
>
> **Q2:**
> In Figure 1 (c), the term “SGD” refers to SGD without momentum. We acknowledge that including comparisons with other memory-efficient methods (e.g., Adam-mini, Muon) in the bar plot would provide a clearer picture. In our revision, we will update the figure to include these baselines. Notably, our method achieves a near memory-optimal footprint—comparable to vanilla SGD (i.e., without momentum)—which is one of the key strengths of SWAN.
>
> **Q3:**
> Indeed, in some settings, this additional momentum before SWAN operations could provide further acceleration. However, there's a trade-off given hardware constraints. For example, consider the task of training a 350M model with 1024 context length under mixed precision on a node of 8XA100 40G GPUs. Including momentum forces the users to use larger gradient accumulation. In contrast, in this scenario, the stateless design of SWAN without momentum allows training with 2X larger batch sizes compared with using momentum. We will clarify these trade-offs in our revision.
>
>
> **Q4:**
> We appreciate the reviewer’s reference to Zhang et al. (2024) and acknowledge its relevance in motivating the need for stateless methods. While we have already cited this paper in our submission, we will expand our discussion in the revision to provide a deeper analysis of how the Hessian perspective discussed in [1] further motivates the design of new stateless optimizers like SWAN.
>
> **Once again, we thank the reviewer for their constructive feedback and insightful questions, which will help us improve the manuscript in the final version.**

---

### Decision · Program_Chairs · 2025-05-01

**Decision:**

Accept (poster)

**Comment:**

This introduces SWAN, a completely stateless optimizer for LLM pre‐training that combines a row‐wise GradNorm step with a matrix‐level GradWhitening step. By preprocessing raw SGD gradients in this way, SWAN matches or surpasses Adam’s token‐efficiency while cutting end‑to‑end memory by ≈ 50 % and eliminating all optimizer states. SWAN represents a significant and practical step toward memory‐efficient LLM training, with strong empirical results and solid theoretical grounding. All the reviewers recommend (weak) accept. The authors should incorporate the discussions and additional experimental results presented during rebuttal into the next revision.